# PCV2 targets cGAS to inhibit type I interferon induction to promote other DNA virus infection

**Zhenyu Wang☉, Jing Chen☉, Xingchen Wu, Dan Ma, Xiaohua Zhang, Ruizhen Li, Cong Han, Haixin Liu, Xiangrui Yin, Qian Du, Dewen Tong\*, Yong Huang💿\***

College of Veterinary Medicine, Northwest A&F University, Yangling, China

☉ These authors contributed equally to this work.
\* dwtong@nwsuaf.edu.cn (DT); yonghuang@nwsuaf.edu.cn (YH)

## Abstract

Viruses use diverse strategies to impair the antiviral immunity of host in order to promote infection and pathogenesis. Herein, we found that PCV2 infection promotes the infection of DNA viruses through inhibiting IFN-β induction *in vivo* and *in vitro*. In the early phase of infection, PCV2 promotes the phosphorylation of cGAS at S278 via activation of PI3K/Akt signaling, which directly silences the catalytic activity of cGAS. Subsequently, phosphorylation of cGAS at S278 can facilitate the K48-linked poly-ubiquitination of cGAS at K389, which can been served as a signal for recognizing by the ubiquitin-binding domain of histone deacetylase 6 (HDAC6), to promote the translocation of K48-ubiquitinated-cGAS from cytosol to autolysosome depending on the deacetylase activity of HDAC6, thereby eventually resulting in a markedly increased cGAS degradation in PCV2 infection-induced autophagic cells relative to Earle's Balanced Salt Solution (EBSS)-induced autophagic cells (a typical starving autophagy). Importantly, we found that PCV2 Cap and its binding protein gC1qR act as predominant regulators to promote porcine cGAS phosphorylation and HDAC6 activation through mediating PI3K/AKT signaling and PKCδ signaling activation. Based on this finding, gC1qR-binding activity deficient PCV2 mutant (PCV2RmA) indeed shows a weakened inhibitory effect on IFN-β induction and a weaker boost effect for other DNA viruses infection compared to wild-type PCV2. Collectively, our findings illuminate a systematic regulation mechanism by which porcine circovirus counteracts the cGAS-STING signaling pathway to inhibit the type I interferon induction and promote DNA virus infection, and identify gC1qR as an important regulator for the immunosuppression induced by PCV2.

## Author summary

PCV2 is well known for its ability to induce immunosuppression in pigs. However, how PCV2 infection interferes cGAS-STING signaling is still poorly understood. Herein, we demonstrate that PCV2 infection can phosphorylate porcine cGAS via gC1qR-mediated PI3K/AKT signaling to silence the catalytic activity of cGAS, while activates PKCδ signaling to promote histone deacetylase 6 (HDAC6) activation depending on the assistance of

**Data Availability Statement:** All relevant data are within the manuscript and its Supporting Information files.

**Funding:** This work was supported by the National Natural Science Foundation of China (grants

31972686 to Y.H. and 31872447 to D.T.). This work was also supported by the Key R&D Program of Shaanxi Province (grants 2018ZDCXL-NY-02-07 and 2020NY-010 to Y.H.) and by Fundamental Research Funds for the Central Universities (grant 2452017023 to Y.H.).The funders had no role in study design, data collection and analysis, decision to publish, or preparation of the manuscript.

**Competing interests:** The authors have declared that no competing interests exist.

gC1qR. Subsequently, phosphorylation of cGAS facilitates the poly-ubiquitination of cGAS, then ubiquitinated-cGAS proteins are recruited and transported to autolysosome by activated HDAC6 depending on its deacetylase activity and ubiquitin-binding function, thereby eventually resulting in the autophagic degradation of cGAS in PCV2-infected cells. This study reveals that PCV2 can inhibit the activation of cGAS signaling pathway through two different mechanisms at different stages of infection and clarifies the internal relationship and cooperation model between these two mechanisms.

## Introduction

Virus infection triggers the host cells to produce type I interferon and proinflammatory cytokines, which is an important defense mechanism for the host to clear viruses. Cyclic GMP-AMP synthase (cGAS) is a cytosolic DNA sensor for recognizing cytosolic DNA derived from microbe or self [1]. Upon DNA triggering, cGAS initiates signal transduction through the synthesis of the second-messenger molecule 2′3′-cGAMP, which diffuses throughout the cell and binds and activates the adaptor protein stimulator of interferon genes (STING). Activated STING oligomerizes and traffics from the endoplasmic reticulum (ER) to the ER-Golgi intermediate compartment (ERGIC), where it recruits and activates TANK-binding kinase 1 (TBK1), thereby activating the transcription factor IRF3 and NF-κB to induce the production of type I interferon and other cytokines [2–4]. The secreted type I interferons stimulate cells to induce the expression of interferon-stimulated genes; the proteins encoded by these genes act as the predominant antiviral agents to inhibit the acute or persistent infection via inhibition of virus replication in different stages.

Not surprisingly, many pathogenic viruses have evolved diverse strategies to disable the type I interferon pathway and evade the host antiviral immune responses [5,6]. Among these strategies, cGAS has been shown to a frequent target of DNA virus antagonism [7]. For example, Kaposi's sarcoma-associated herpesvirus (KSHV) tegument protein ORF52 inhibits cGAS enzymatic activity and cGAMP production via binding to DNA and cGAS, or targeting cGAS-DNA phase separation [8,9]; likewise, tegument protein VP22 from alpha-herpesvirus also can restrict cGAS-DNA phase separation or interact with cGAS to mediate immune evasion [9,10]; Human cytomegalovirus (HCMV) protein UL31 interacts directly with cGAS to disassociate DNA from cGAS, thus inhibiting cGAS enzymatic function [11]. Herpes simplex virus 1 (HSV-1) tegument protein UL37 deamidates human and mouse cGAS to impair the ability of cGAS to catalyze cGAMP synthesis [12], and UL41 has been reported to directly degrade cGAS mRNA [13]. RNA viruses also have been shown to counteract cGAS-STING pathway. For example, Dengue virus NS2B protease cofactor targets the cGAS for lysosomal degradation [14,15], and ZIKV infection promotes the degradation of cGAS via activation of caspase-1 [16]. In addition, we noted that distinct types of polyubiquitination of cGAS have been identified in these studies. For example, K27-linked polyubiquitination of cGAS at Lys173and Lys384 by RNF185 promotes its enzymatic activity during HSV-1 infection [17]; K48-linked polyubiquitination of cGAS at Lys414 promotes its p62-dependent autophagic degradation during HSV-1 infection by an unknown E3 ligase [18]; K48-linked polyubiquitination of cGAS at Lys271 and Lys464 contributes to homeostasis of cGAS for proper initiation and attenuation of an innate immune response in the different conditions [19]. In these studies, several molecules have been identified that regulate the enzymatic activity and degradation of cGAS, however, the exact mechanisms of action remain unclear. For example, it is unclear whether some specific molecules participate in spatially transportation of the

ploy-ubiquitinated cGAS from cytosol to the autolysosome and whether other types of post-translation modifications of cGAS are associated with the polyubiquitinin of cGAS. All of the above-mentioned questions remain to be answered.

Porcine circoviruses (PCV) are non-enveloped small viruses with single-stranded circular DNA genomes (~1.7–2.0 kb) within genus Circovirus in the family Circoviridae [20,21]. Four types of circoviruses have been reported in pigs [22]. PCV1 is nonpathogenic; PCV2 is considered the main pathogen of porcine circovirus-associated diseases (PCVAD), including respiratory and enteric disease, reproductive failure, porcine dermatitis and nephropathy syndrome (PDNS), and PCV2-systemic disease [23]. PCV3 and PCV4 are newly discovered in recent years [24]. PCV2 infection induces the immunosuppression of host, and leads to susceptibility of infected pigs to multiple pathogens [25], yet how PCV2 infection interferes cGAS-STING signaling to promote the infection of themselves and other DNA viruses is still poorly understood. In this study, we investigated the mechanisms by which PCV2 temporally coordinates to regulate the enzyme activity, poly-ubiquitination, and degradation of cGAS, and identified the predominant viral proteins and host proteins that responsible for regulating enzyme activity, poly-ubiquitination, transportation, and degradation of cGAS during PCV2 infection. This study therefore gains a better understanding of how PCV2 infection suppresses type I interferon production to increase viral replication and pathogenesis.

## Results

### PCV2 inhibits the activation of type I IFN signaling to facilitate the infection of other DNA viruses by reducing cGAMP production

Previous studies have reported that PCV2 infection leads to immunosuppression in pigs and increased susceptibility to other pathogens [25–27]. Epidemiological investigation showed that the infection rate and pathogenic rate of porcine parvovirus (PPV) and porcine pseudorabies virus (PRV) were significantly increased in PCV2 positive pigs (Fig 1A and 1B); the PPV and PRV loads in native PCV2 positive pigs are higher than that in native PCV2 negative pigs (Fig 1C). Consistent with this observation, PCV2 infection markedly promoted PRV and PPV replication in cellular experiments (Fig 1D). Upon PRV or PPV challenge, serum IFN-β, tissue IFN-β and interferon-stimulated genes (IFIT1, CXCL10) mRNA levels were more significantly upregulated in mock- and PCV1-infected pigs than that in PCV2-infected pigs (Fig 1E and 1F). Similar results were observed in samples of interferon-stimulated DNA (ISD) (S1A and S1B Fig), as well as in HT-DNA-induced IFN-β on VSV-GFP replication (Fig 1G). Importantly, the inhibitory effects on IFN-β induction by PCV2 was dose and time-dependent (S2A and S2B Fig), yet independent of type I interferon receptor (S2C and S2D Fig), suggesting that the inhibitory effect of PCV2 on type I interferon induction by other DNA viruses is closely related to the PCV2 infection level.

To figure out how PCV2 infection influences the type I interferon expression, we tested and compared cGAMP production, phosphorylated TBK1 levels, and the phosphorylation and translocation of IRF3 in mock and PCV2-infected cells. PRV infection or ISD stimulation did significantly induce cGAMP production, TBK1 phosphorylation, and IRF3 phosphorylation and translocation in mock- and PCV2-infected cells, while we noted that cGAMP, phosphorylated TBK1, phosphorylated IRF3, and intranuclear IRF3 were remarkably decreased in PCV2-infected cells relative to them in Mock-infected cells at 24 or 48 h. p.i, and further decreased with the increase of PCV2 infection dose or time (Fig 1H, 1I, 1J and 1K). A substantial decrease in cGAMP production was observed in 5 multiplicity of infection (MOI) of PCV2-infected cells at 48 h and 72 h (S2E Fig). Taken together, these data demonstrated that

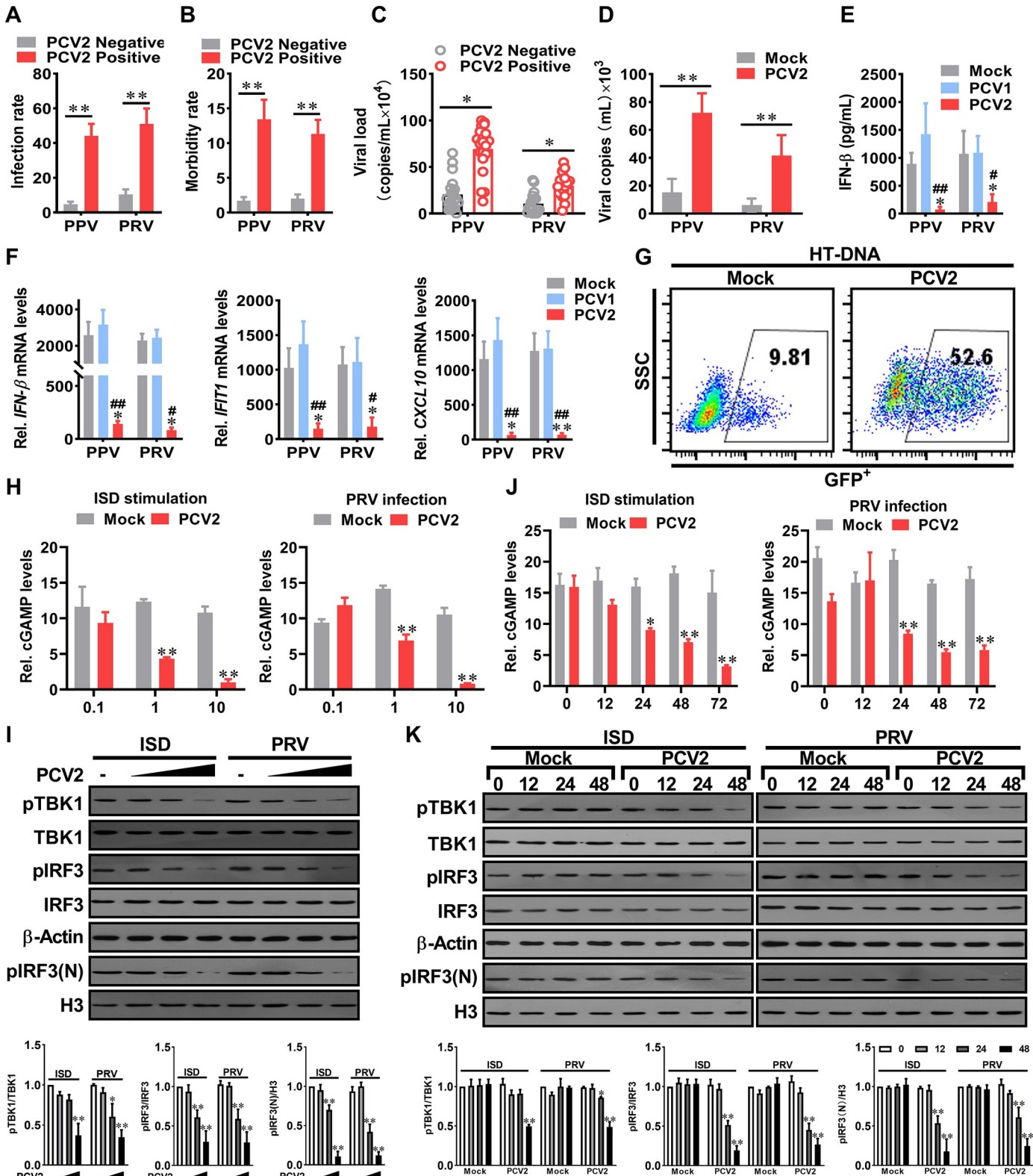

**Fig 1. PCV2 inhibits the activation of type I IFN signaling to facilitates the infection of other DNA viruses by reducing cGAMP production. (A, B)** The infection rate and morbidity of porcine parvovirus (PPV) or porcine pseudorabies virus (PRV) were compared between PCV2 positive pigs and PCV2 negative pigs. The native pig herds (n = 452) were separated into PCV2 positive group (n = 271) and PCV2 negative group (n = 181), then the infection rates (A) and the morbidity rates (B) of PPV or PRV were further analyzed in these two groups. **(C)** The PPV and PRV loads in native PCV2 positive pigs are higher than that in

native PCV2 negative pigs. The viral copy numbers of PPV and PRV in the serum of native PCV2 positive pigs (n = 20 per group) and native PCV2 negative pigs (n = 17 per group) were measured by qPCR. $*$ $P < 0.05$, $**$ $P < 0.01$ (compared with PCV2 negative pigs). **(D)** PPV and PRV replication levels in PCV2-infected cells are higher than that in mock infection cells. The PK-15 cells were infected with mock or PCV2 for 48 h, then were further infected with PPV or PRV, and the relative viral titers were measured by $TCID_{50}$. $*$ $P < 0.05$, $**$ $P < 0.01$(compared with mock infection). **(E, F)** PCV2 inhibits PPV- or PRV-induced IFN-β production and response. The piglets were infected by PCV1 ($4 \times 10^5$ $TCID_{50}$), PCV2 ($4 \times 10^5$ $TCID_{50}$), or mock (same volume of medium) for 1 week, respectively, and then challenged with $10^5$ $TCID_{50}$ PPV or $10^5$ $TCID_{50}$ PRV for another 24 h. The serum IFN-β of the infected piglets were measured by ELISA (E); IFN-β, IFIT1, and CXCL10 mRNA levels in lung tissues were determined by qPCR (F). $*$ $P < 0.05$, $**$ $P < 0.01$ (compared with mock infection); $\#$ $P < 0.05$, $\#\#$ $P < 0.01$ (compared with PCV1 infection). **(G)** Comparison of VSV-GFP replication in PK-15 cells pretreated with the cell supernatants from HT-DNA-stimulated mock-infected cells or HT-DNA-stimulated PCV2-infected cells. GFP positive cells were measured by flow cytometry. **(H-K)** PK-15 cells were infected with different doses (0.1, 1, 10 MOI) of PCV2 for 24 h (H, I), or infected with PCV2 (MOI = 1) for the indicated time (J, K), and then the relative cGAMP production levels (H, J), the levels of p-TBK1, p-IRF3, and nucleoprotein pIRF3 (pIRF3(N)) at 6 h following ISD stimulation or PRV infection were determined by report assay and western blotting, respectively. $*$ $P < 0.05$, $**$ $P < 0.01$(compared with mock infection).

PCV2 infection inhibits other DNA viruses-induced type I IFN signaling by reduction of cGAMP production, which promotes the infection of DNA viruses.

## PCV2 infection results in cGAS degradation through activation of autophagy-lysosome pathway

Given that PCV2 infection inhibited the production of cGAMP induced by other DNA viruses or ISD, thus we proposed that PCV2 infection might inhibit the catalytic activity of cGAS or reduce the protein level of cGAS. Indeed, cGAS protein levels considerably decreased with increased PCV2 infection times (Fig 2A, upper), whereas the abundance of cGAS mRNA did not change with increased viral levels (Fig 2A, lower), suggesting that PCV2 infection promotes cGAS degradation. We also observed a reduction of cGAS protein in the cells inoculated with high doses of ultraviolet-inactivated PCV2, even though the dead virus showed a weaker effect than the living virus (Fig 2B), suggesting that PCV2 induced-cGAS degradation did not absolutely depend on the process of viral replication, but it is closely related to the level of PCV2 in cells.

To evaluate the mechanism of cGAS degradation during PCV2 infection, we firstly tested which one was employed to promote the degradation of cGAS in two major protein degradation systems (ubiquitin-proteasome pathway and autophagy-lysosome pathway) by PCV2. Results showed that the lysosomal acidification inhibitor chloroquine (CQ) effectively prevented the reduction of cGAS protein in PCV2-infected cells, but the proteasome inhibitor MG132 did not (Fig 2C). In parallel experiments, either knockdown of ATG5 or treatment with the autophagic-sequestration inhibitor 3-methyladenine (3-MA) also alleviated PCV2-induced cGAS degradation through inhibition of the formation of autophagy (Fig 2D and 2E), indicating that the degradation of cGAS was governed by the autophagy-lysosome pathway in PCV2-infected cells. Confocal microscopy analysis further supported this conclusion showing that more cGAS proteins were co-localized with LC3 in PCV2-infected cells when compared the cells treated with CQ to the cells treated with DMSO, but this co-localization of cGAS with LC3 almost disappeared in the cells pretreated with autophagosome-formation inhibitor 3-methyladenine (3-MA) (Fig 2F and 2G). Interestingly, we noted a phenomenon that blocking the formation of autophagosome simultaneously reduced the replication of the virus, which is consistent with the findings in previous studies[28]. We next used ultraviolet-inactivated PCV2 to infect cells in the presence of different concentrations of CQ. Results showed that the levels of cGAS were increased with the increase of CQ concentration, together with that of p62 (an autophagy flux marker), in the presence of the same doses of inactivated PCV2 (Fig 2H). This result further confirmed that the formation of autophagy flux is required for the reduction of cGAS during PCV2 infection. Taken together, these results demonstrated that

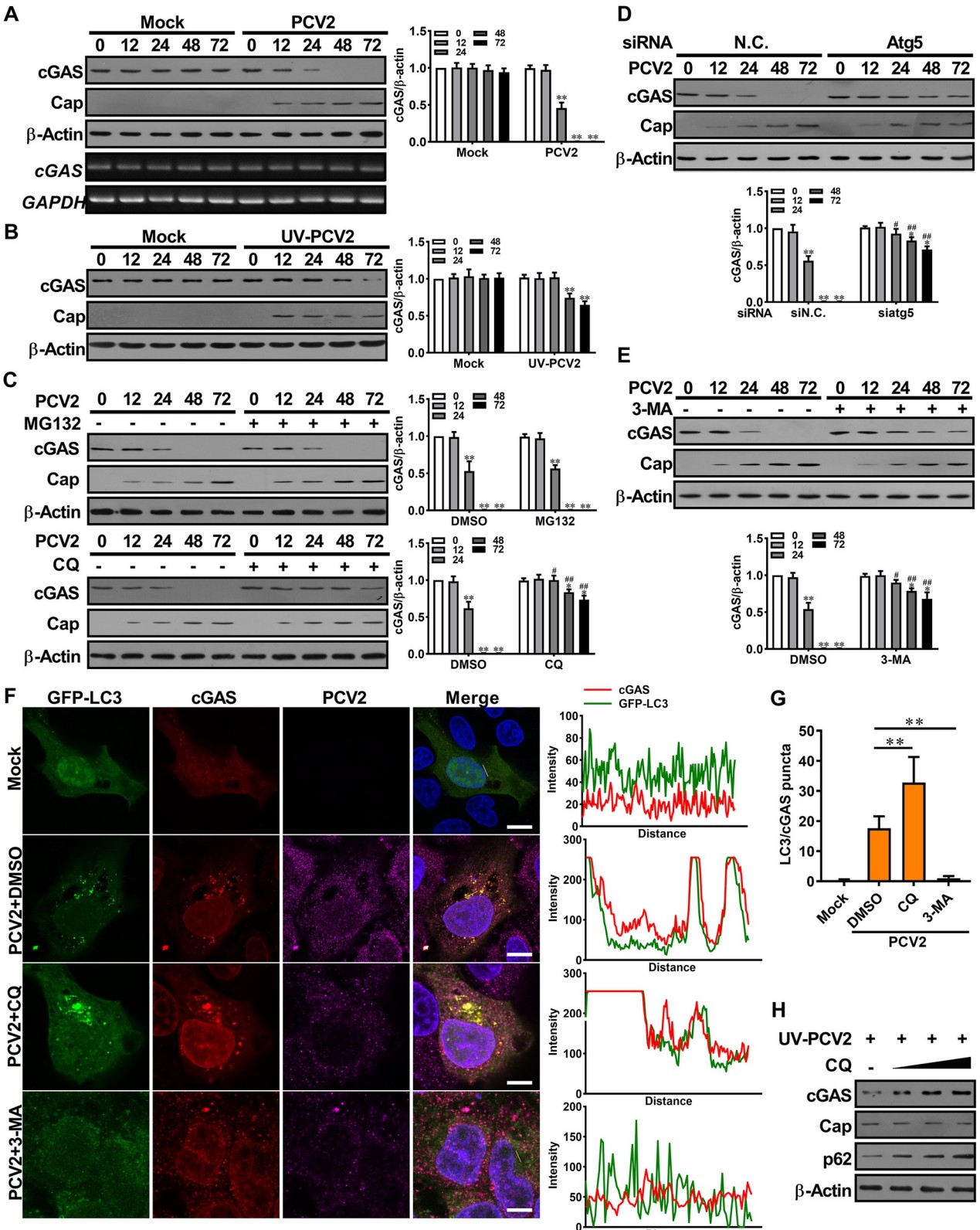

**Fig 2. PCV2 infection results in cGAS degradation through activation of autophagy-lysosome pathway. (A)** PCV2 infection induces porcine cGAS reduction. PK-15 cells were infected with PCV2 (MOI = 5) or mock for the indicated time, and then the protein and mRNA levels of porcine cGAS were determined by western blotting (upper panel) and RT-PCR (lower panel). **(B)** Ultraviolet-inactivated PCV2 induces porcine cGAS

reduction. PK-15 cells were inoculated with UV-PCV2 (MOI = 5) or mock for the indicated time, and then the levels of porcine cGAS and PCV2 capsid were determined by western blotting. **(C)** Chloroquine (CQ) can effectively prevent the reduction of cGAS protein in PCV2-infected cells. PK-15 cells were pretreated with 10 μM MG132 (proteasome inhibitor) or 20 μM CQ (autophagy inhibitor) for 8 h, then infected with PCV2 (MOI = 5) for indicated times to detect cGAS levels. ** $P < 0.01$ (compared with infection at 0 h), # $P < 0.05$, ## $P < 0.01$ (compared with DMSO treatment in indicated same time). **(D, E)** knockdown of Atg5 or autophagic-sequestration inhibitor 3-MA treatment alleviates PCV2-induced cGAS degradation. PK-15 cells were transfected with Atg5 specific siRNA (siAtg5) or siRNA negative control (siN.C.) for 24 h, or treated with 3-MA (5 mM) for 8 h before PCV2 infection, and then the levels of porcine cGAS were determined at the indicated times by western blotting. * $P < 0.05$, ** $P < 0.01$ (compared with infection at 0 h), # $P < 0.05$, ## $P < 0.01$ (compared with siRNA negative control or DMSO treatment in indicated same time) **(F, G)** cGAS proteins are co-localized with LC3 in PCV2-infected cells. PK-15 cells transfected with GFP-LC3 were infected with PCV2 in the presence of DMSO, CQ or 3-MA along, followed by confocal observation (F, Left). The co-localization signals of targeted proteins were analyzed as intensity profiles of indicated proteins along the plotted lines by Image J line scan analysis (F, Right). The number of yellow puncta was counted and compared (G). Scale bar, 10 μm. ** $P < 0.01$ (compared with DMSO treatment). **(H)** PK-15 cells were pretreated with different doses (10, 20, 40 μM) of CQ for 8 h before UV-PCV2 inoculation, and then the levels of porcine cGAS, PCV2 Cap and p62 were determined at 48 hpi.

PCV2 infection induces the autophagic degradation of cGAS by promoting translocation of cGAS to the autophagosome.

## PCV2 infection enhances the K48-linked poly-ubiquitination of porcine cGAS at K389 and facilitates the interaction of cGAS with p62 for autophagic degradation

Ubiquitination is a common biochemical modification in the living organism that regulates many cellular physiological responses, which generates linkage-specific degrons on substrates destined for degradation [29,30]. In PCV2-infected cells (MOI = 1), the ubiquitination of porcine cGAS was detected, and ubiquitinated cGAS was accumulated in the presence of bafilomycin A1 (Baf) (Fig 3A). Since different types of poly-ubiquitination, such as K6-, K11-, K27-, K29-, K33-, K48-, or K63-linked ubiquitination are known to be implicated in regulating protein fate in cells [31], we set out to analyze which type of ubiquitination was involved in PCV2-moadultion of the porcine cGAS function. Results revealed that porcine cGAS was predominantly ubiquitinated with K48 linkage, but not with other types of ubiquitin chains at the late phase of PCV2 infection (Fig 3B and 3C). Meanwhile, the K48-linked poly-ubiquitination and degradation of porcine cGAS were also detected in Earle's Balanced Salt Solution (EBSS)-treated cells (these cells exhibit typical starving autophagy), which was further accumulated in the presence of Baf or CQ (S3A Fig), suggesting that porcine cGAS was ubiquitinated and degraded in EBSS-induced autophagy. Notably, in EBSS-treated cells, PCV2 infection further increased the K48-linked poly-ubiquitination of cGAS, resulting in more cGAS degradation relative to that of mock infection (Fig 3D). Overall, these results indicated that PCV2 infection induces K48-ubiquitination of porcine cGAS, leading to its degradation via autophagy.

To further explore the mechanism of PCV2-induced cGAS ubiquitination and degradation, we predicted the potential ubiquitination sites in porcine cGAS through alignment of similar conserved sites between human cGAS and porcine cGAS (S3B Fig), following the method reported in previous study [32], and mapped potential ubiquitination sites in porcine cGAS by constructing a series of lysine (K)-to-arginine (R) substitution in 8 potential ubiquitination sites. Compared to wild-type or mutants at other lysine sites, the K48-linked ubiquitination of K389R cGAS mutant was abrogated in both PCV2-infected cells and EBSS-treated cells (Figs 3E and 3F and S3C), thus K389R cGAS mutant was not degraded in both PCV2-infected cells and EBSS-treated cells (Figs 3F and S3D), suggesting that the lysine residue at position 389 of porcine cGAS is required for its K48-linked poly-ubiquitination whatever PCV2 infection or EBSS treatment. As we expected, PCV2 infection also promoted the interaction of wild-type cGAS with p62 but failed to promote the interaction of K389R cGAS mutants with p62 (Figs 3G and 3H and S3E and S3F). These data together demonstrated that PCV2 infection enhances

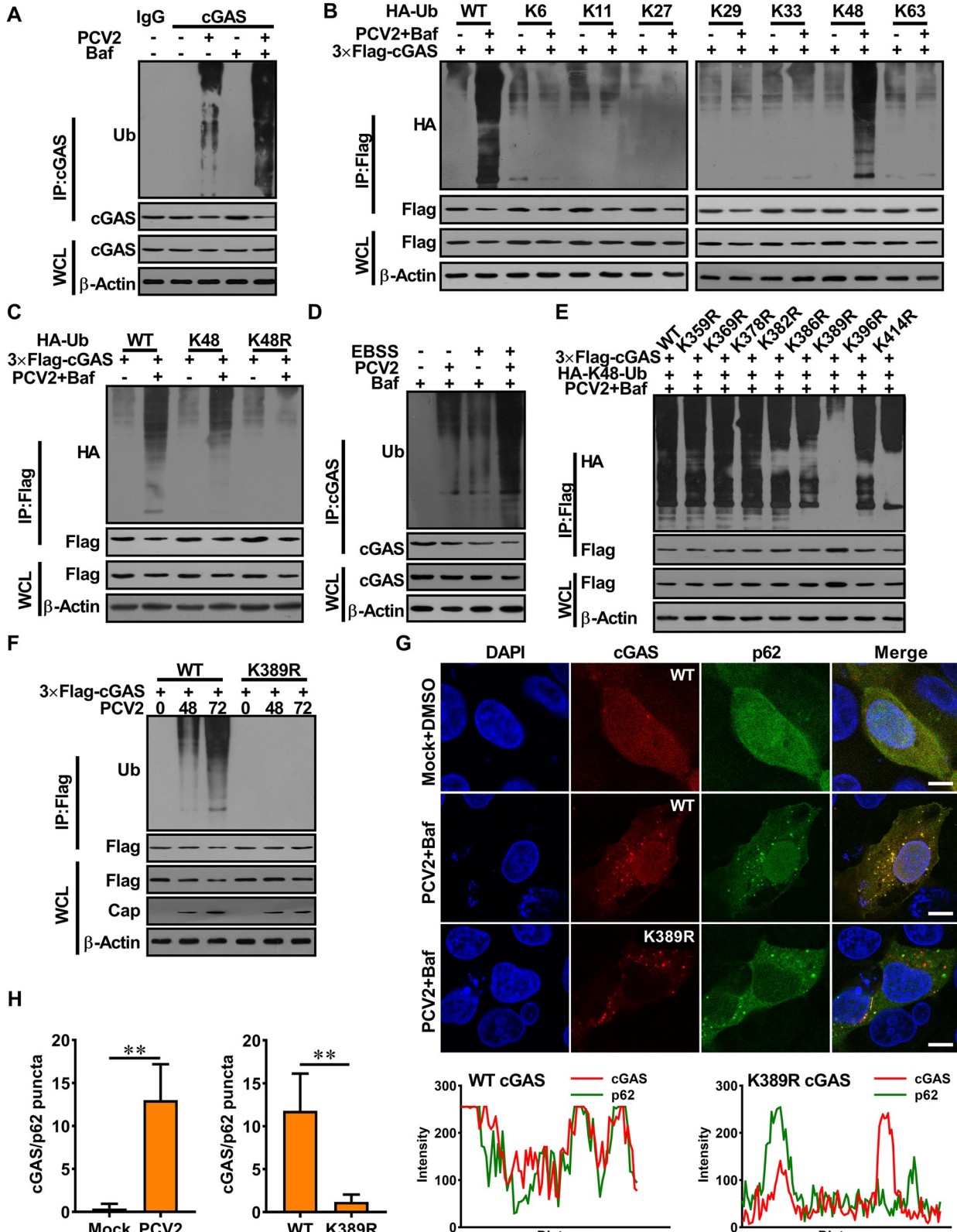

**Fig 3. PCV2 infection enhances the K48-linked poly-ubiquitination of porcine cGAS at K389 and facilitates the interaction of cGAS with p62 for autophagic degradation. (A)** PCV2 infection enhanced the ubiquitination of porcine cGAS. PK-15 cells were infected with PCV2

(MOI = 5) along with or without Baf for 48 h. Cell lysates were analyzed by immunoprecipitated with anti-porcine cGAS antibody, and ubiquitinated cGAS proteins were immunoblotted using anti-ubiquitin antibodies. **(B, C)** Porcine cGAS was mainly ubiquitinated with K48 linkage in PCV2-infected cells. PK-15 cells were transfected with different HA-Ub constructs as indicated, then infected with PCV2 in the presence or absence of Baf. Cell lysates were immunoprecipitated with anti-Flag antibody and immunoblotted with anti-HA antibody. **(D)** PK-15 cells were treated with EBSS together with or without PCV2 infection (MOI = 5) in the presence of BAF for 48 h. The poly-ubiquitination levels and protein levels of cGAS were analyzed. **(E)** The lysine 389 of porcine cGAS is required for its K48-ubiquitination induced by PCV2 infection. PK-15 cells were transfected with cGAS mutant expression vectors as indicated, then cells were infected with PCV2 (MOI = 5) in the presence of Baf for 48 h, the poly-ubiquitination levels of cGAS were analyzed. **(F)** Mutation of the lysine 389 of porcine cGAS can block poly-ubiquitination and degradation of porcine cGAS induced by PCV2. PK-15 cells were transfected with plasmids as indicated, then infected with PCV2 (MOI = 5) for the indicated time. Protein lysates were immunoprecipitated with anti-Flag antibody and immunoblotted with anti-Ub antibody. **(G, H)** Poly-ubiquitination of porcine cGAS at K389 is required for the interaction of cGAS with p62. The cGAS$^{-/-}$ PK-15 cells expressed Flag-cGAS, Flag-cGAS (K389R), then infected with Mock or PCV2 along with Baf (DMSO as a control). Cells were subjected to confocal assay to observe the co-localization of cGAS and p62. The co-localization signals of targeted proteins were analyzed as intensity profiles of indicated proteins along the plotted lines by Image J line scan analysis (G). Scale bar, 10 μm. Statistics of the puncta formation by cGAS-p62 in the indicated samples (H). ** $P$ < 0.01 (compared with Mock or WT).

the K48-linked poly-ubiquitination of porcine cGAS at K389, which is critical for facilitating the interaction of cGAS with p62 and the autophagic degradation of cGAS.

## PCV2 infection activates HDAC6 to mediate the transport of poly-ubiquitin cGAS to autolysosome

Now that PCV2 appears to have a strong ability to promote K48-ubiquitination and degradation of porcine cGAS relative to EBSS treatment, while we also noted that the site of K48-linked poly-ubiquitination of porcine cGAS is shared between the EBSS and PCV2 infection. Thus, we hypothesized that some specific molecules are involved in the K48-ubiquitination of cGAS, or the transport and degradation of poly-ubiquitin cGAS during PCV2 infection. In the cells infected with a lower multiplicity of PCV2, ubiquitinated cGAS proteins were gradually accumulated with the prolonged infection times in the presence of autolysosome inhibitor bafilomycin A1 (Baf) (Fig 4A). HDAC6 is a key molecule in promoting the interaction of ubiquitinated proteins with the autophagy adaptor p62, functioning as a ubiquitin-binding and transport platforms to recruit ubiquitinated proteins for further degradation via autophagy [30]. During PCV2 infection, despite no significant changes in HDAC6 expression levels, the deacetylase activity of HDAC6 was markedly enhanced, which led to an apparent reduction in the levels of Ac-tubulin (the natural active substrate of HDAC6) (Figs 4B, lower panel, and S4A). Knockdown of HDAC6 markedly decreased the degradation of cGAS, did not affect the K48-ubiquitination levels of cGAS in PCV2-infected cells in the presence of Baf (Figs 4B and S4B), yet markedly increased the K48-ubiquitination levels of cGAS in PCV2-infected cells in the absence of Baf (S4C Fig). However, knockdown HDAC6 did not significantly affect the degradation of cGAS in EBSS-treated cells (S4D Fig). Meanwhile, we noted that the deacetylase activity of HDAC6 was not markedly enhanced upon EBSS treatment (S4E Fig). These data suggested that EBSS-induced cGAS autophagic degradation does not depend on the activation of HDAC6, while PCV2-induced cGAS degradation is dependent on the activation of HDAC6.

To further identified the exact roles of HDAC6 in mediating cGAS autophagic degradation, we examined a potential interaction of cGAS with endogenous HDAC6. In PCV2-infected cells, ubiquitinated cGAS were interacted with HDAC6 and p62 (Figs 4C and 4D and S4F and S4G), whereas K389R mutated cGAS were not able to interacted with HDAC6 and p62 (Fig 4D and 4E). Knockdown of HDAC6 significantly decreased the interaction of ubiquitinated cGAS with p62, but ubiquitinated cGAS did not decrease in HDAC6 knockdown cells (Fig 4F). Similarly, in HDAC6$^{-/-}$ PK-15 cells, the interaction of ubiquitinated cGAS with p62 significantly decreased, which could be reversed by wild-type HDAC6 reconstitution, but not by a

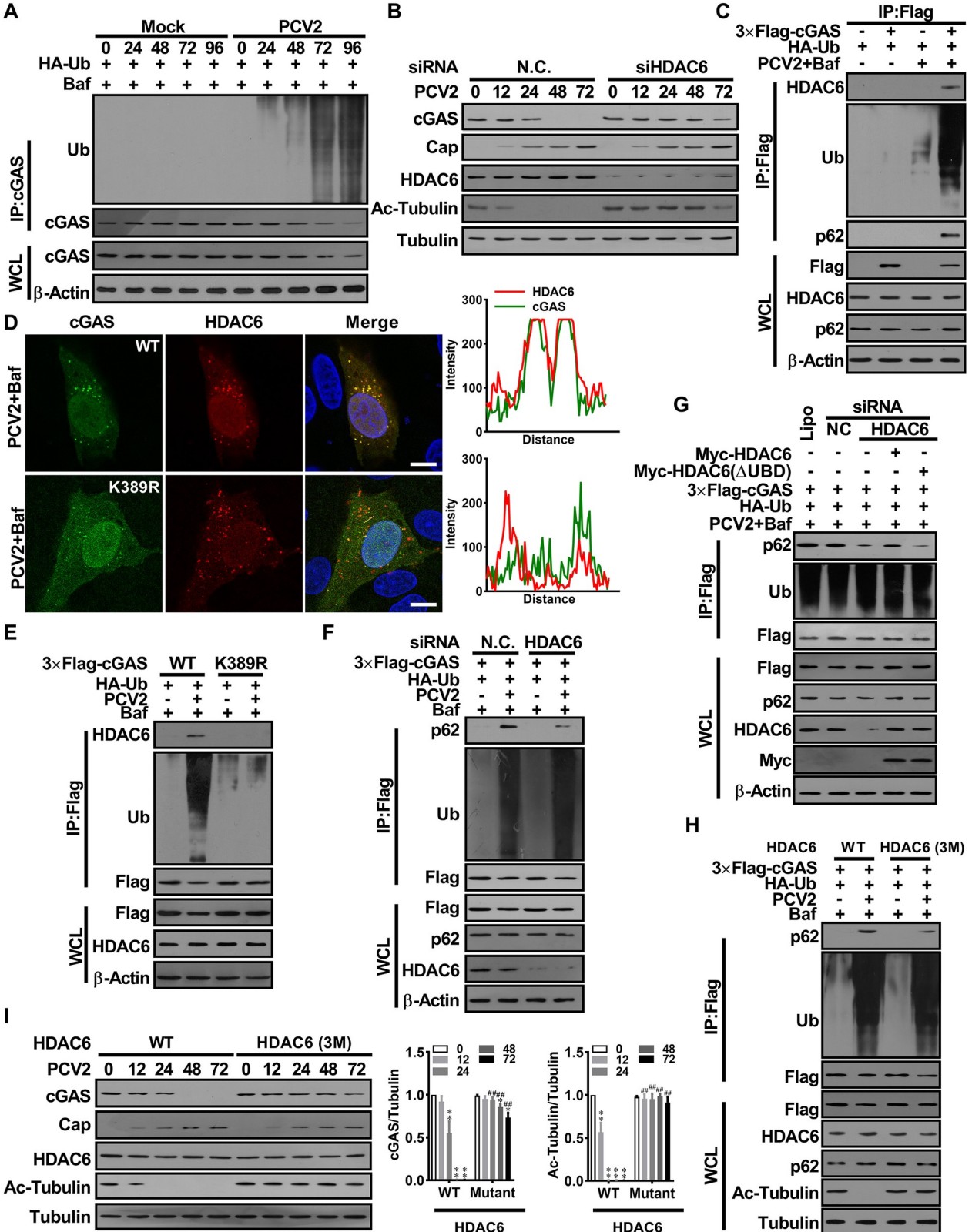

**Fig 4. PCV2 infection activates HDAC6 to mediate the transport of poly-ubiquitin cGAS to the lysosome via HDAC6 mediation.** **(A)** PK-15 cells were infected with PCV2 (MOI = 1) for the indicated times and the poly-ubiquitination levels and protein levels of cGAS were analyzed. **(B)**

PK-15 cells were transfected with HDAC6 specific siRNA (siHDAC6) or siRNA negative control (siN.C.) for 24 h before PCV2 infection (MOI = 5), and then the levels of porcine cGAS, PCV2 Cap, HDAC6, Ac-tubulin were determined by western blotting at the indicated time post-infection. **(C)** PK-15 cells were transfected plasmids indicated, then infected with PCV2 (MOI = 5) in the presence of Baf for 48 h. the interaction of ubiquitinated cGAS with HDAC6 and p62 was analyzed. **(D)** The cGAS$^{-/-}$ PK-15 cells transfected with Flag-cGAS, Flag-cGAS (K389R) expression constructs were infected with PCV2 in the presence of Baf. The colocalization of porcine cGAS and HDAC6 were observed under confocal microscopy. Scale bar, 10 μm. The co-localization signals of targeted proteins were analyzed as intensity profiles of indicated proteins along the plotted lines by Image J line scan analysis. **(E)** PK-15 cells were transfected expression vectors as indicated, then infected with PCV2 in the presence of Baf for 48 h. Subsequently, the ubiquitination levels of cGAS and cGAS (K389R), and their ability to bind HDAC6 were measured. **(F)** PK-15 cells were transfected with HDAC6 specific siRNA (siHDAC6) or siRNA negative control (siN.C.) and indicated plasmids for 24 h. Then these cells were infected with PCV2 (MOI = 5) or mock in the presence of Baf for 48 h; the interaction of ubiquitinated cGAS with P62 was analyzed. **(G)** PK-15 cells were transfected with HDAC6 specific siRNA (siHDAC6) or siRNA negative control (siN.C.) and indicated plasmids, meanwhile, Myc-HDAC6 (FL), Myc-HDAC6 (ΔUBD) were reconstituted in the cells transfected siHDAC6. Then, these cells were infected with PCV2 (MOI = 5) for 48 h, and the interaction of ubiquitinated cGAS with p62 was analyzed in these cells. **(H)** The HDAC6$^{-/-}$ PK-15 cells were transfected wild-type HDAC6 (WT) or mutated HDAC6-3M (D538A/D608A/S609A) for 24 h, then infected with PCV2 (MOI = 5) for another 48 h, and the interaction of ubiquitinated cGAS with p62 was analyzed. **(I)** The HDAC6$^{-/-}$ PK-15 cells transfected wild-type HDAC6 (WT) or mutated HDAC6-3M for 24 h, then infected with PCV2 (MOI = 5) for the indicated time, and then the levels of porcine cGAS, PCV2 Cap, and Ac-Tubulin were determined by western blotting. $^{*}$ $P < 0.05$, $^{**}$ $P < 0.01$ (compared with infection at 0 h), $^{\#}$ $P < 0.05$, $^{\#\#}$ $P < 0.01$ (compared with wild type HDAC6 group in indicated same time).

mutant HDAC6 that loss of its ubiquitin-binding domain (Fig 4G), suggesting that the ubiquitin-binding domain of HDAC6 play a critical role in mediating the interaction of ubiquitinated cGAS with p62. Interestingly, inhibition of deacetylase activity of HDAC6 by its specific inhibitor (Tubastatin A, Tub A) significantly decreased the interaction of ubiquitinated cGAS with p62 and led to a decreased cGAS degradation in PCV2-infected cells (S4H and S4I Fig). However, inhibition of the deacetylase activity of HDAC6 did not affect the interaction of ubiquitinated cGAS with p62 and cGAS degradation in EBSS-treated cells (S4J and S4K Fig). To further confirm this finding, we detected the interaction of ubiquitinated cGAS with p62 and cGAS levels in the HDAC6 deficient cells that rescued with mutated porcine HDAC6 (D538A/D608A/D609A, named as HDAC6-3M), which abolished the deacetylation activity of HDAC6 in these cells, while significantly decreased the interaction of ubiquitinated cGAS with p62 and the degradation of cGAS during PCV2 infection (Fig 4H and 4I). Taken together, these results demonstrated that PCV2 infection activates HDAC6 to mediate the transport of poly-ubiquitinated cGAS to autolysosome.

## PCV2 infection induces phosphorylation of porcine cGAS at Ser278 to negatively regulate cGAS enzymatic activity

Given that PCV2 infection markedly reduces the levels of porcine cGAS within cells through autophagic degradation, we proposed that blocking the autophagic degradation of cGAS might be able to efficiently weaken the inhibitory effects of PCV2 on type I interferon production. Thus, we investigated whether or not autophagic-sequestration inhibitor 3-methyladenine (3-MA) and lysosomal acidification inhibitor chloroquine (CQ) could effectively alleviate the suppression of PCV2 on cGAMP production and IFN-β induction. However, upon PRV challenge or ISD stimulation, we found no difference for cGAMP and IFN-β mRNA levels between CQ/3-MA-treated PCV2-infected cells and control DMSO treated PCV2-infected cells, even though the levels of cGAS protein was markedly higher in CQ/3-MA-treated PCV2-infected cells than that in control DMSO-treated PCV2-infected cells (Fig 5A and 5B). Likewise, mutation of K389 site of porcine cGAS and inhibition of HDAC6 activity that prevents the K48-ubiquitination of cGAS and the transport of ubiquitinated cGAS to increase the levels of cGAS in PCV2-infected cells, but these interventions were not sufficient to weaken the inhibitory effects of PCV2 on cGAMP induction (Fig 5C and 5D). These results suggested that the block of cGAS autophagic degradation is unable to restore the function of cGAS.

The aforementioned data led us to hypothesize that PCV2 infection may inhibit cGAS enzymatic activity directly. Previously studies have reported that the phosphorylation of human

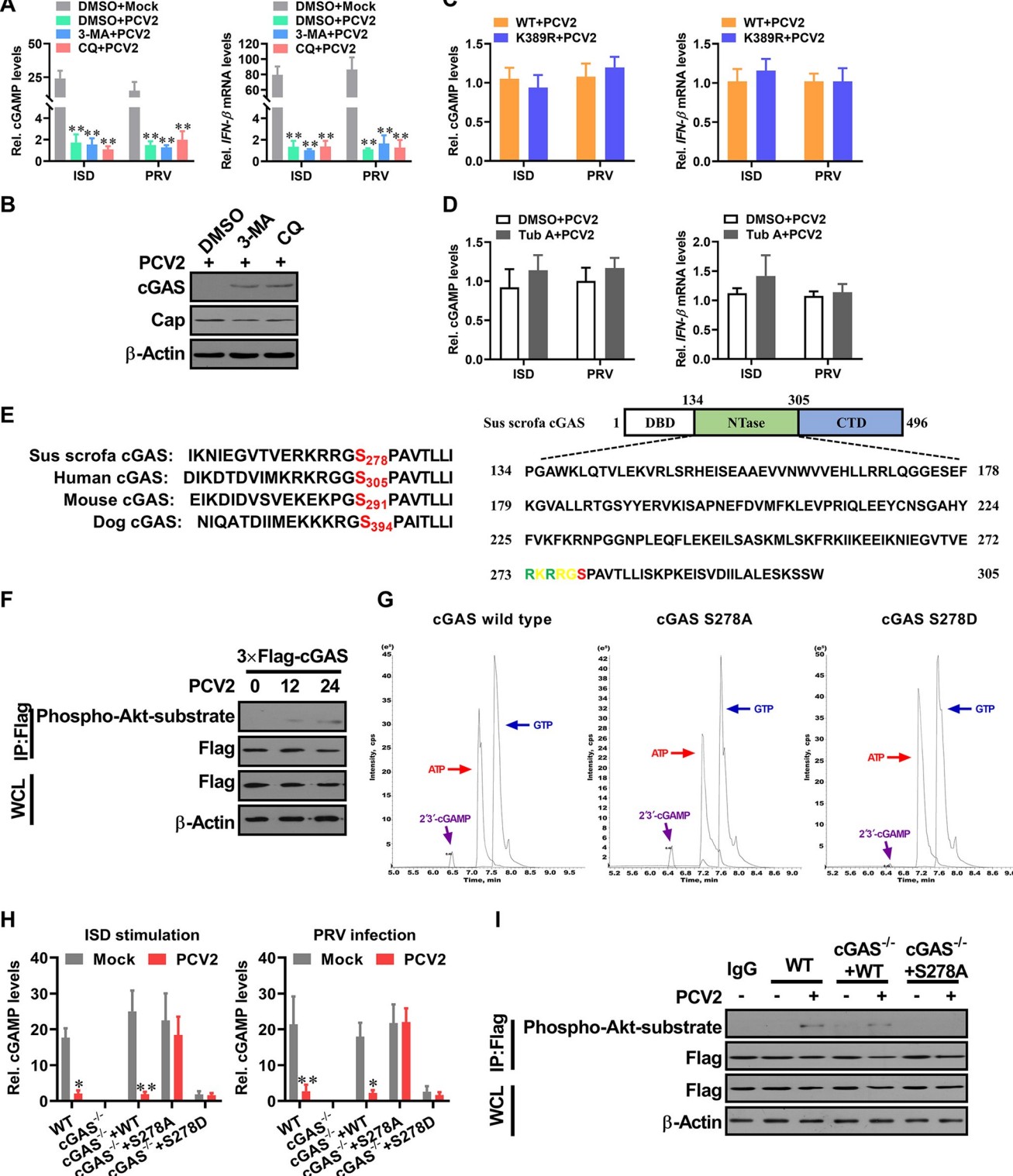

**Fig 5. PCV2 induces phosphorylation of porcine cGAS at Ser278 to negatively regulate cGAS enzymatic activity. (A, B)** Suppression of autophagic flux cannot effectively improve the induction of cGAMP and IFN-β in PCV2-infected cells. PK-15 cells pretreated with 3-MA or CQ were infected with PCV2 (MOI = 5) for 48 h, and then the relative cGAMP production levels and IFN-β mRNA levels at 6 h following ISD stimulation or PRV infection were determined by report assay and qPCR respectively (A). ** $P < 0.01$ (compared with mock infection). The levels of porcine cGAS and PCV2 capsid were determined by western blotting (B). **(C)** The cGAS[-/-] PK-15 cells were reconstituted with the wild type cGAS, mutant cGAS (K389R), respectively, then

infected with PCV2 (MOI = 5) for 48 h to detect the relative cGAMP and IFN-β mRNA production levels at 6 h following ISD stimulation or PRV infection. **(D)** PK-15 cells pretreated with Tub A (DMSO as control) were infected with PCV2 (MOI = 5) for 48 h, then the relative cGAMP and IFN-β mRNA production levels were determined at 6 h following ISD stimulation or PRV infection. **(E)** Sequence comparison of the cGAS phosphorylation site from the indicated species (upper panel). The potential phosphorylated peptides in the NTase motif of porcine cGAS are highlighted in green and red (lower panel). **(F)** PK-15 cells were infected with mock or PCV2 for indicated times, and the phosphorylation level of cGAS at the S278 site was detected by western blotting using a rabbit anti-Phospho-Akt Substrate monoclonal antibody that can recognize the motif (RRGS*$_{278}$) of porcine cGAS. **(G)** The phosphorylation of porcine cGAS at Ser278 exhibits a weakened enzymatic activity in *in vitro* assay. LC-MS analysis of cGAMP production from an *in vitro* cGAMP synthesis assay. Small molecules were extracted from in vitro tubes which contained the same doses of wild-type cGAS, phosphorylation-resistant S278A mutant, and phosphomimetic S278D mutant for analysis of cGAMP isomers by tandem mass spectrometry. **(H)** The cGAS$^{-/-}$ PK-15 cells reconstituted with the WT cGAS, or cGAS mutant S278A, or cGAS mutant S278D were infected with mock or PCV2 in the presence of Baf for 12 h, and then the relative cGAMP production levels were determined at 6 h following ISD stimulation or PRV infection. * $P < 0.05$, ** $P < 0.01$ (compared with mock infection). **(I)** The cGAS$^{-/-}$ PK-15 cells reconstituted with the WT cGAS, or cGAS mutant S278A were infected with mock or PCV2 for 12 h, and then the phosphorylation level of cGAS at the S278 site was detected by Immunoprecipitation.

cGAS at S305 suppresses its enzymatic activity [33]. In the cGAS proteins derived from different species, the catalytic core of the NTase domain contain a highly conservative motif (R/KXR/KXX*S/T; X, any amino acid) (Figs 5E and S5A). According to this prediction, we used an antibody that could specifically recognize the phosphorylated epitope of this motif in porcine cGAS. The results showed that porcine cGAS was phosphorylated at the S278 site at the early phase of PCV2 infection (Fig 5F). To further confirm the role of S278 phosphorylation in the modulation of porcine cGAS activity, we compared the enzyme activities of wild-type porcine cGAS, phosphorylation-resistant mutant S278A, and phosphomimetic S278D mutant via *in vitro* catalytic activity assay. As expected, purified cGAS S278A mutant proteins showed similar cGAMP production activity as well as wild-type cGAS *in vitro* (Fig 5G), whereas the same dose of phosphomimetic S278D mutant was not able to catalyze the substrate to produce cGAMP in the same reaction condition (Figs 5G and S5B and S5C). To explore the role of serine 278 phosphorylation in the regulation of porcine cGAS activity, we compared the production capability of cGAMP in PCV2-infected cGAS$^{-/-}$ PK-15 cells that rescued with either FLAG-tagged wild-type porcine cGAS, FLAG-tagged phosphorylation-resistant porcine cGAS mutant (S278A), or phosphomimetic S278D mutant. Upon DNA stimulation or PRV infection, cGAMP production was markedly decreased in PCV2-infected wild-type PK-15 cells and cGAS$^{-/-}$ PK-15 cells that rescued with wild-type porcine cGAS relative to the same cells without PCV2 infection, but cGAMP production was not found to be reduced in PCV2-infected cGAS$^{-/-}$ PK-15 cells that rescued with porcine cGAS mutant (S278A) relative to the same cells without PCV2 infection in the presence of Baf (Figs 5H and S5D); in both wild-type PK-15 cells and cGAS$^{-/-}$ PK-15 cells that rescued with wild-type porcine cGAS, the phosphorylation of serine 278 was detected upon PCV2 infection, whereas this phosphorylation was disappeared in PCV2-infected cGAS$^{-/-}$ PK-15 cells that rescued with porcine cGAS mutant (S278A) (Fig 5I). Notably, in cGAS$^{-/-}$ PK-15 cells that were rescued with phosphomimetic S278D mutant, both ISD and PRV were not able to significantly induce cGAMP production whenever PCV2 infection or not (Fig 5H). These results demonstrated that phosphorylation of porcine cGAS at serine 278 is critical for modulation of its catalytic activity, and suggested that PCV2 infection can modulate the enzyme activity of porcine cGAS through inducing the phosphorylation of porcine cGAS at serine 278.

## The phosphorylation of porcine cGAS at S278 facilitates the K48-linked poly-ubiquitination and degradation of porcine cGAS during PCV2 infection

To investigate which signaling was responsible for the phosphorylation of porcine cGAS at S278 during PCV2 infection, we examined the potential roles of PI3K/Akt, p38-mitogen-

activated protein kinase (MAPK), JNK, extracellular signal-regulated kinase (ERK), and AMPK signaling pathways, which have been identified to be activated by PCV2 infection. The result showed that Akt specific inhibitor or siRNA markedly increased cGAMP production in PCV2-infected cells when these cells were further stimulated by ISD or PRV, whereas p38-MAPK, JNK, ERK, and AMPK specific inhibitors or siRNAs did not (Figs 6A and S6A). Consistently, the phosphorylation of porcine cGAS at serine 278 was also decreased in the PCV2-infected cells treated with Akt specific inhibitor or siRNA (Figs 6B and S6B). However, we could not detect phosphorylation of porcine cGAS at serine 278 treated with Akt specific agonist in PK-15 cells (S6C Fig). Interestingly, we found that the poly-ubiquitination levels of cGAS were decreased in the PCV2-infected cells treated with Akt specific inhibitor or siRNA compared to the cells that treated with DMSO or siRNA negative control, but did not change in the PCV2-infected cells treated with p38-MAPK, JNK, ERK, and AMPK specific inhibitors or siRNAs (Figs 6C and S6D). The protein levels of cGAS were also increased in the PCV2-infected cells that treated with Akt specific inhibitor or siRNA compared to the cells treated with DMSO or siRNA negative control, but did not change in the PCV2-infected cells treated with p38-MAPK, JNK, and ERK specific inhibitors or siRNAs (Figs 6D and S6E) In the PCV2-infected cells treated with AMPK specific inhibitor or siRNA, however, the protein level of cGAS was increased despite the poly-ubiquitination level of cGAS did not decrease (Figs 6D and S6E). These results demonstrated that the phosphorylation of porcine cGAS at S278 is regulated by Akt signaling during PCV2 infection and suggested that K48-linked poly-ubiquitination and degradation of cGAS are associated with the phosphorylation of cGAS at S278 in the process of PCV2 infection.

Next, we confirmed whether the phosphorylation of cGAS at S278 is required for K48-linked poly-ubiquitination and degradation of cGAS in the process of PCV2 infection. In cGAS$^{-/-}$ PK-15 cells, transfected WT cGAS was phosphorylated at S278 and further K48 poly-ubiquitinated at K389 (Fig 6E); the transfected cGAS S278A mutant was able to be further K48 poly-ubiquitinated at K389 upon PCV2 infection, but the level of K48 poly-ubiquitinated cGAS was markedly reduced (Fig 6E). However, the S278D phosphomimetic cGAS retained the potential to be K48 poly-ubiquitinated at K389 as well as wild-type cGAS upon PCV2 infection (Fig 6E). Importantly, upon EBSS treatment, porcine cGAS (S278A) mutant showed similar levels of K48 poly-ubiquitinated at K389 as was the WT porcine cGAS (S6F Fig), suggesting that the phosphorylation of cGAS at S278 is not required for K48-linked poly-ubiquitination and degradation of porcine cGAS, but the phosphorylation of porcine cGAS at S278 indeed can facilitate the K48-linked poly-ubiquitination of porcine cGAS during PCV2 infection. Consistent with this poly-ubiquitination result, cGAS protein bearing the single S278A mutation displayed a decreased binding with HDAC6 and a decreased colocalization with p62 in PCV2-infected cells (Figs 6F and 6G and S6G). Taken together, these data indicated that the phosphorylation of porcine cGAS at S278 facilitates the K48-linked poly-ubiquitination and degradation of porcine cGAS during PCV2 infection.

## PCV2 Cap plays a predominant role in promoting porcine cGAS phosphorylation and HDAC6 activation depending on gC1qR and/or PKCδ

To make clear which protein of PCV2 plays a predominant role in promoting the phosphorylation and polyubiquitination of cGAS, we firstly detected the phosphorylation of porcine cGAS in PK-15 cells that were infected with PCV1 mutants that replaced ORF1 or ORF2 in PCV1 backbone by PCV2 ORF1 or ORF2 (named PCV1-Rep2 and PCV1-Cap2) and PCV2 mutants that replaced ORF1 or ORF2 in PCV2 backbone by PCV1 ORF1 or ORF2 (named

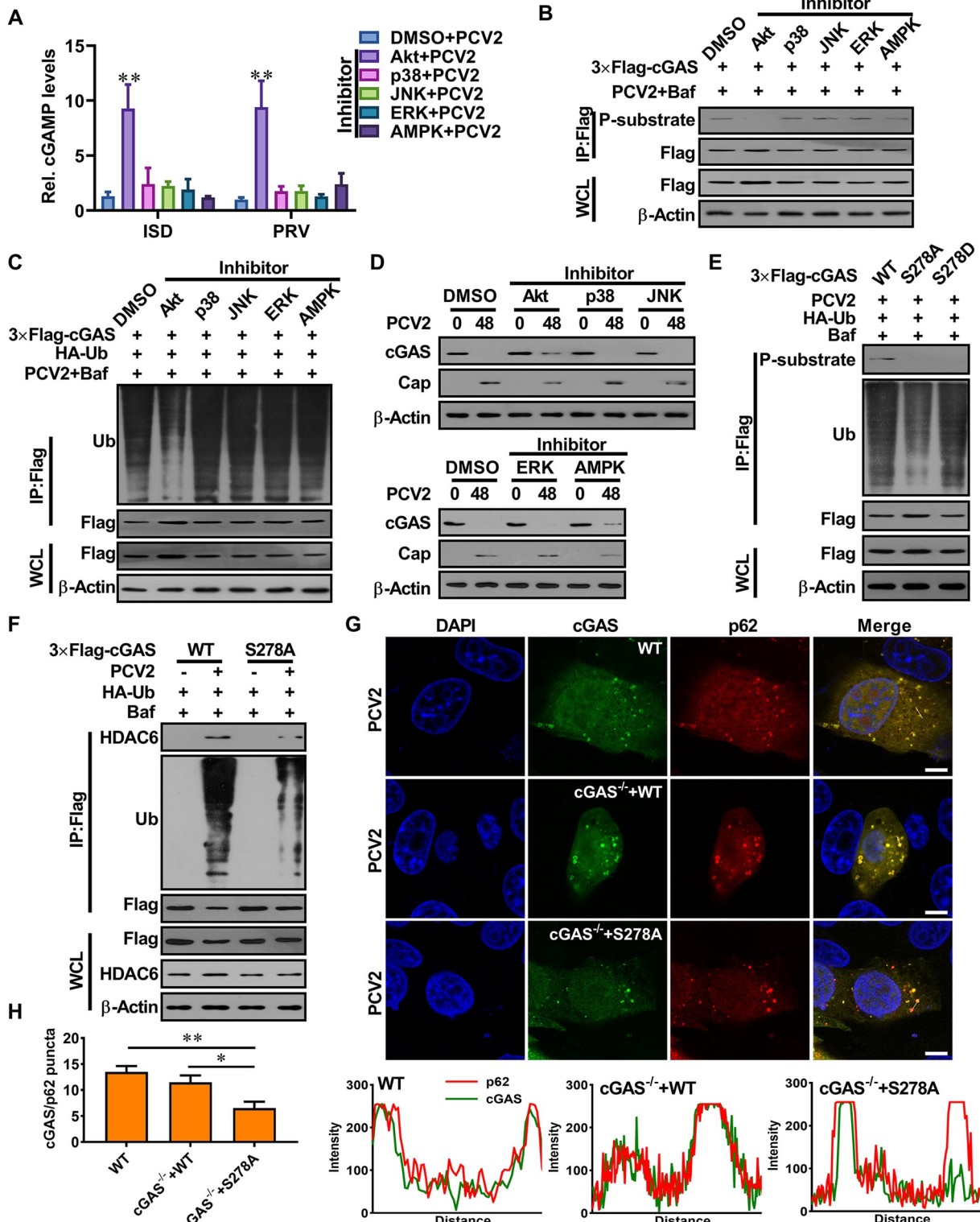

**Fig 6. The phosphorylation of porcine cGAS at Ser278 facilitates the K48-linked poly-ubiquitination and degradation of porcine cGAS during PCV2 infection. (A-B)** Akt signaling negatively regulates cGAS-mediated cGAMP production in the PCV2-infected cells via phosphorylation of porcine cGAS at S278. PK-15 cells pretreated with indicated inhibitors were infected with PCV2 (MOI = 5) for 12 h, and then the relative cGAMP production levels were determined at 6 h following ISD stimulation or PRV infection (A); the phosphorylation level of cGAS at the S278 site was detected by western blotting (B). $^*$ $P < 0.05$, $^{**}$ $P < 0.01$ (compared with DMSO/mock infection). **(C, D)** The

phosphorylation of cGAS facilitates the ubiquitination and degradation of cGAS during PCV2 infection. PK-15 cells transfected with indicated expression constructs were treated with indicated inhibitors and then infected with PCV2 (MOI = 5) in the presence or absence of Baf to detect the poly-ubiquitination levels and protein levels of cGAS. **(E)** The cGAS$^{-/-}$ PK-15 cells were reconstituted with the WT cGAS, or S278A mutant, or S278D mutant, then infected with PCV2 (MOI = 5) to detect the poly-ubiquitination levels and phosphorylation levels of cGAS in these cells. **(F)** The cGAS$^{-/-}$ PK-15 cells were reconstituted with WT cGAS, or S278A mutant, then infected with PCV2 (MOI = 5) or mock to detect the interaction of ubiquitinated cGAS with HDAC6. **(G)** The wild-type PK-15 cells, and cGAS$^{-/-}$ PK-15 cells were reconstituted with WT cGAS or S278A mutant, then infected with PCV2 to observe the colocalization of cGAS and p62. The co-localization signals of targeted proteins were analyzed as intensity profiles of indicated proteins along the plotted lines by Image J line scan analysis. Scale bar, 10 μm.

PCV2-Rep1 and PCV2-Cap1) which were constructed in our previous studies (Fig 7A) [27,34]. Results showed that porcine cGAS was phosphorylated at the S278 site in the cells infected with PCV2, PCV2-Rep1 or PCV1-Cap2, but not in the cells infected with PCV1, PCV1-Rep2 or PCV2-Cap1 (Fig 7B). Consistently, phosphorylation of porcine cGAS at serine 278 was also detected in the cells infected with recombinant adenoviruses expressing PCV2 Cap (rAd-Cap), but not in the cells infected with rAd-Rep or rAd-Blank (Fig 7C). Furthermore, the poly-ubiquitinated at K389 was able to be detected in the cells infected with rAd-Cap, but not in the cells infected with rAd-Rep or rAd-Blank (Fig 7D). These results demonstrated that PCV2 Cap protein may play a predominant role in promoting phosphorylation and poly-ubiquitination of cGAS. Next, we test the roles of PCV2 Cap-binding protein gC1qR in promoting phosphorylation of cGAS. As we expected, the phosphorylation of porcine cGAS at S278 was not detected in gC1qR deficient cells upon PCV2 infection (Fig 7E), suggesting that porcine cGAS phosphorylation is dependent on gC1qR during PCV2 infection.

To further explore which protein of PCV2 plays a predominant role in promoting HDAC6 activation, we measured the changes of HDAC6 activity by monitoring the levels of acetylated α-tubulin (the natural active substrate of HDAC6) in cells infected with different PCV mutants. The result showed that the levels of Ac-tubulin were apparently reduced in PCV2-, PCV2-Rep1- or PCV1-Cap2-infected cells with the increase of infection time, but not in PCV1-, PCV1-Rep2- or PCV2-Cap1-infected cells, indicated that the deacetylase activity of HDAC6 was markedly enhanced in the cells expression PCV2 Cap protein, despite all of them did not show a significant change in HDAC6 protein levels (Fig 7F). Next, we test the roles of PCV2 Cap-binding protein gC1qR in promoting HDAC6 activation. The level of Ac-tubulin increased in gC1qR deficient cells compared with wild-type cells upon PCV2 infection (Fig 7G), suggesting that PCV2 Cap together with gC1qR to promote HDAC6 activation in PCV2-infected cells.

A previous study had reported that SeV infection induced the activation of PKCα which promoted its interaction with HDAC6 and enhanced its deacetylation activity in a phosphorylation-dependent manner [35]. However, in our previous study, we found that PCV2 infection can phosphorylate PKC-δ, but not PKC-α [36]. Thus, we further explored whether PKCδ was involved in HDAC6 activation in PCV2-infected cells. Firstly, we observed that HDAC6 indeed interacted with PKCδ in PCV2-infected cells (Fig 7H); HDAC6-mediated tubulin deacetylation was markedly decreased in the PCV2-infected cells when treated with an inhibitor of PKCδ isoform (Rottlerin) or transfected with PKCδ specific siRNA (siPKCδ) (Fig 7I and 7J). Considering our previous findings that treated with PKCδ isoform inhibitor (Rottlerin) or knockdown of PKCδ can decrease progeny virus production of PCV2 through inhibiting nuclear egress of virion [37], we further detected the effects of exogenous expression of PCV2 Cap on the activity of HDAC6, and the roles of PKCδ during this process. The results showed that expression of PCV2 Cap could promote HDAC6 activation, while treatment with PKCδ isoform inhibitor (Rottlerin) or knockdown of PKCδ could suppress HDAC6 activation in Cap-expressing cells (Fig 7K and 7L), These results indicated that PKCδ is a key regulator in promoting the activation of HDAC6 in PCV2 infected cells.

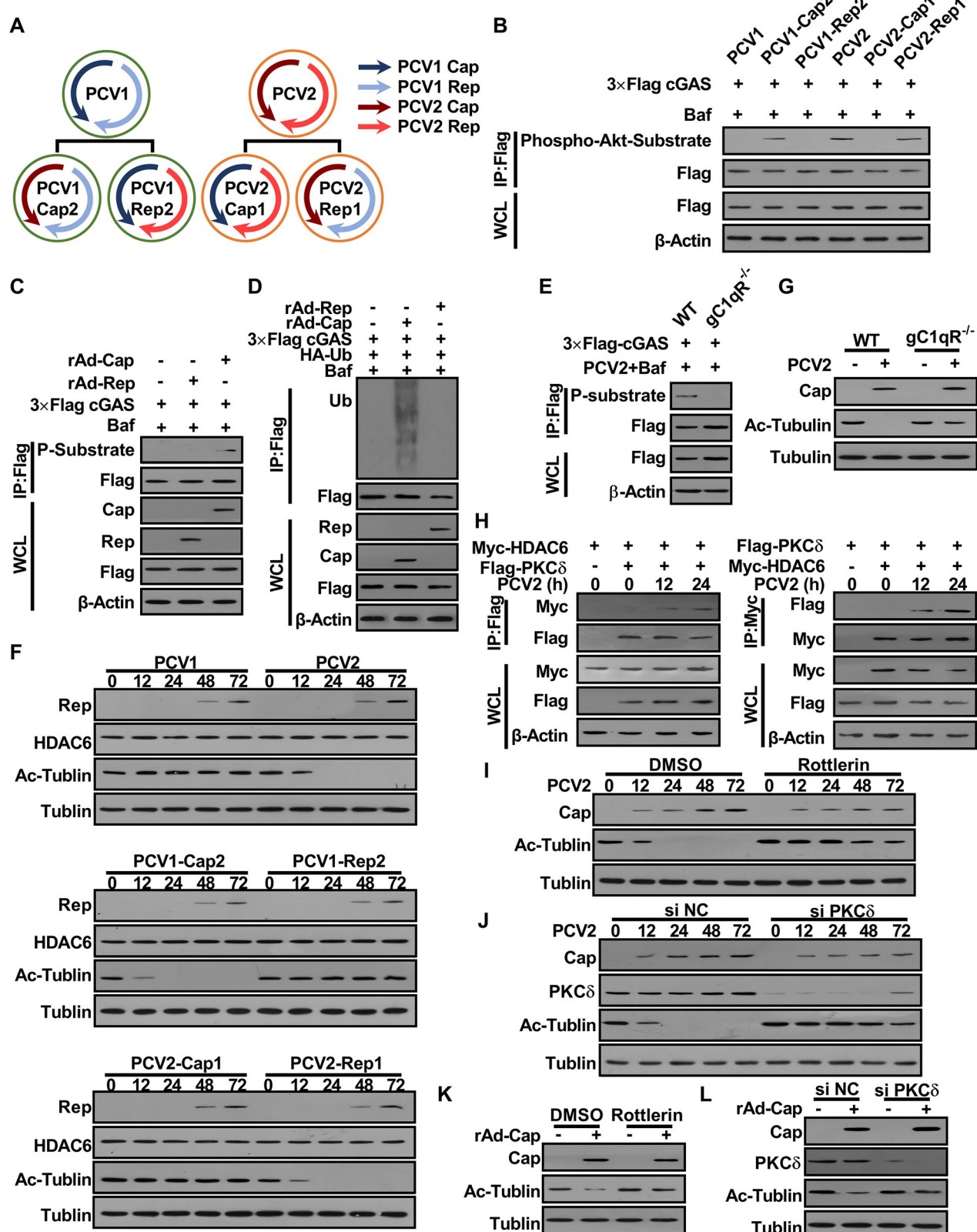

**Fig 7. PCV2 Cap plays a predominant role in promoting porcine cGAS phosphorylation and HDAC6 activation depending on gC1qR and/or PKCδ. (A)** Model of construction of PCV mutants. **(B-D)** Cap is a critical regulator of cGAS phosphorylation. PK-15 cells were infected with PCV

mutants (MOI = 5) for 12 h, and then the phosphorylation level of cGAS at the S278 site was detected by western blotting (B). PK-15 cells were infected with rAd-Blank (MOI = 100), rAd-Rep (MOI = 100) and rAd-Cap (MOI = 100) for 24 h, then the phosphorylation (C) and the poly-ubiquitination levels (D) of cGAS were detected by immunoprecipitation. (E) gC1qR$^{-/-}$ PK-15 cells and wild type PK-15 cells were infected with PCV2 (MOI = 5) for 12 h, then the phosphorylation of cGAS was detected by immunoprecipitation. (F) Cap is a critical regulator of HDAC6 activation. PK-15 cells were inoculated with PCV mutants (MOI = 5) for the indicated time, and then the levels of HDAC6 and Ac-tubulin were determined by western blotting. (G) gC1qR$^{-/-}$ PK-15 cells and wild type PK-15 cells were infected with PCV2 (MOI = 5) for 48 h, then the levels of Ac-Tubulin detected by western blotting. (H) Interaction of PKCδ with HDAC6. PK-15 cells were transfected with Myc-HDAC6 and Flag-PKCδ for 24 h. Then these cells were infected with PCV2 (MOI = 5) for indicated time; the interaction of HDAC6 with PKCδ was analyzed. (I) PK-15 cells were pretreated with Rottlerin and infected with PCV2 (MOI = 5) for the indicated time, and then the levels of Ac-Tubulin were determined by western blotting. (J) PK-15 cells were transfected with PKCδ specific siRNA (siPKCδ) or siRNA negative control (siN.C.) for 24 h and then infected with PCV2 (MOI = 5) for the indicated time, followed by western blotting detection of the levels of Ac-Tubulin. (K, L) PK-15 cells were pretreated with Rottlerin (K), or transfected with PKCδ specific siRNA (siPKCδ) or siRNA negative control (siN.C.) (L), and then infected with rAd-Cap (100 MOI) for 24 h, followed by western blotting detection of the levels of Ac-Tubulin.

## gC1qR-binding activity deficient PCV2 mutant (PCV2RmA) shows a weakened inhibitory effect on IFN-β induction and a weaker boost effect for other DNA viruses

gC1qR, as a multiligand-binding, multicompartmental, and multifunctional protein, interacts with PCV2 Cap during PCV2 infection [38]. Previous studies from our lab and others have shown that PCV2 infection promotes anti-inflammatory response via gC1qR-mediated PI3K/Akt signaling activation [27,39]. Herein, either PPV- or PRV-induced cGAMP and IFN-β mRNA was significantly higher in PCV2-infected gC1qR-deficient cells than that in PCV2-infected wild-type cells (S7A and S7B Fig); likewise, either PPV- or PRV-induced cGAMP and IFN-β mRNA in the cells infected with rAd-Cap were significantly lower than that in the cells infected with rAd-Rep or rAd-Blank (S7C and S7D Fig). These data indicated that Cap and its binding protein gC1qR play predominant roles in the suppression of type I interferon production in PCV2-infected cells.

PCV2 infection promotes phosphorylation of cGAS in wild-type cells, but not in gC1qR-deficient cells (S7E Fig). Notably, activation of Akt signaling using Akt agonist (SC79) was not able to induce the phosphorylation of porcine cGAS at serine 278 (S7E Fig). Based on this function of gC1qR, we investigated whether blocking the interaction of gC1qR with PCV2 Cap can weaken the inhibitory ability of PCV2 on type I interferon production, Thus, we evaluated the inhibitory ability of PCV2 mutant PCV2RmA (in which mutated Cap loses the ability to bind gC1qR) to other DNA-induced cGAMP and IFN-β. As expected, upon ISD stimulation, PRV or PPV infection, both IFN-β and cGAMP production in PCV2RmA-infected cells were significantly higher than that in wild-type PCV2-infected cells (Fig 8A and 8B). Furthermore, we tested whether simultaneous rescuing cGAS level and activity through knockdown of HDAC6 and inhibition of AKT signaling could completely remove the inhibitory effects of PCV2 on IFN-β induction. Results showed that PPV or PRV-induced IFN-β expression did increase a litter bit in PCV2-infected cells when HDAC6 and AKT were simultaneously silenced or inhibited, but IFN-β expression was still lower in PCV2-infected HDAC6 knockdown cells than that in PCV2RmA-infected HDAC6 knockdown cells in the presence of AKT inhibitor (LY294002) (Fig 8B). However, cGAMP production was markedly increased in PCV2-infected cells when HDAC6 and AKT were simultaneously silenced or inhibited, and showed a similar level between PCV2-infected HDAC6 knockdown cells and PCV2RmA-infected HDAC6 knockdown cells in the presence of AKT inhibitor (LY294002) (Fig 8C). These data suggested that simultaneous rescuing cGAS level and activity through knockdown of HDAC6 and inhibition of AKT signaling indeed remove the inhibitory effects of PCV2 on cGAMP production, but cannot effectively remove the inhibitory effects of PCV2 on IFN-β induction. Notably, PCV2RmA, a gC1qR binding activity deficient PCV2 mutant, did not

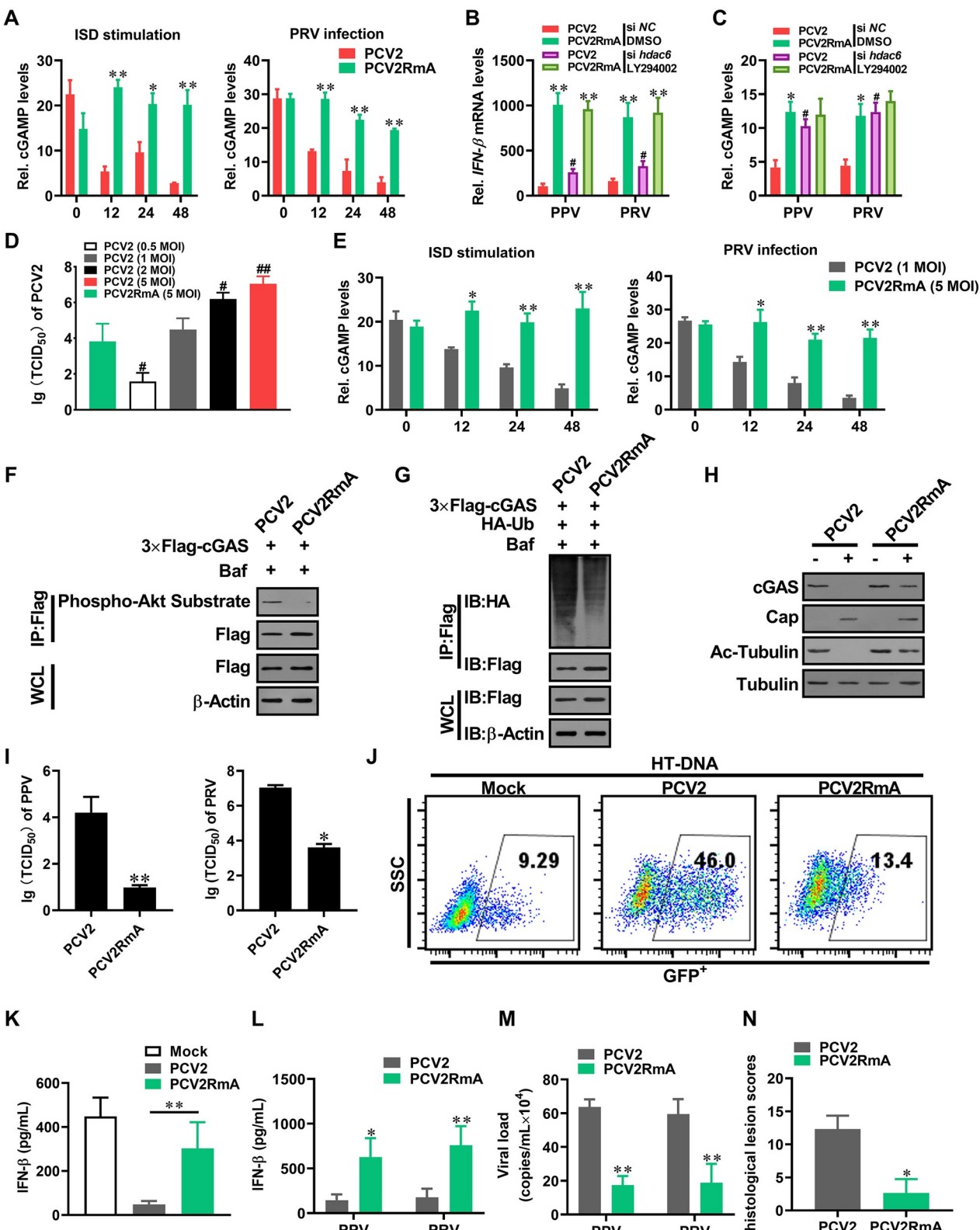

**Fig 8. gC1qR-binding activity deficient PCV2 mutant (PCV2RmA) shows a weakened inhibitory effect on IFN-β induction and a weaker boost effect for other DNA viruses. (A)** PCV2RmA exhibits a lower inhibitory effect on other DNA-induced cGAMP compared with wild type PCV2. PK-15 cells were infected with wild type PCV2 (MOI = 5) or PCV2RmA (MOI = 5) for the indicated time, and then the relative cGAMP production

levels were determined at 6 h following ISD stimulation or PRV infection. $^*$ $P < 0.05$, $^{**}$ $P < 0.01$ (compared with PCV2 infection). **(B, C)** PK-15 cells were transfected with HDAC6 specific siRNA (siHDAC6) or control siRNA (siNC) for 24 h, then treated with Akt inhibitor or DMSO for 6 h, followed by PCV2 (MOI = 5) or PCV2RmA (MOI = 5) infection, and then the IFN-β mRNA levels or cGAMP production were determined at 6 h following PRV or PPV infection. $^*$ $P < 0.05$, $^{**}$ $P < 0.01$ (compared with PCV2 infection in indicated same condition). $^{\#}$ $P < 0.05$ (compared with the cells that transfected siNC and treated with DMSO in same infection). **(D)** PK-15 cells were infected with PCV2 at 0.5~5 MOI or PCV2RmA at 5 MOI, and progeny virion production was determined by $TCID_{50}$ at 48 h post-infection. $^{\#}$ $P < 0.05$, $^{\#\#}$ $P < 0.01$ (compared with 5 MOI PCV2RmA-infected PK-15 cells). **(E)** PK-15 cells were infected with wild type PCV2 (MOI = 1) or PCV2RmA (MOI = 5) for the indicated time, and then the relative cGAMP production levels and IFN-β mRNA levels were determined at 6 h following ISD stimulation or PRV and PPV infection. $^*$ $P < 0.05$, $^{**}$ $P < 0.01$ (compared with PCV2 infection). **(F-H)** PCV2RmA exhibits a lower induction capability in the phosphorylation, poly-ubiquitination, and degradation of porcine cGAS compared with wild type PCV2. PK-15 cells were infected with wild type PCV2 (MOI = 1) or PCV2RmA (MOI = 5) for 12 h, the phosphorylation of cGAS was detected by immunoprecipitation (F), PK-15 cells were infected with wild type PCV2 (MOI = 1) or PCV2RmA (MOI = 5) for 48 h, then poly-ubiquitination (G), and protein levels of cGAS and Ac-tubulin levels (H) were detected by immunoprecipitation. **(I-K)** PCV2RmA is a weak strain relative to PCV2 in promotion of DNA virus replication. PK-15 cells were infected with PCV2 (MOI = 1) or PCV2RmA (MOI = 5) for 48 h, then were further infected with PPV or PRV, and the relative viral titers were measured by $TCID_{50}$ (I). $^*$ $P < 0.05$, $^{**}$ $P < 0.01$. (J) Comparison of VSV-GFP replication in PK-15 cells pretreated with cell supernatants from HT-DNA-stimulated mock-infected cells, HT-DNA-stimulated PCV2-infected cells, or HT-DNA-stimulated PCV2RmA-infected cells. GFP positive cells were measured by flow cytometry. (K) IFN-β levels were measured by ELISA. $^{**}$ $P < 0.01$. **(L-N)** PCV2RmA alleviating PPV- or PRV-induced pathological changes. The piglets (n = 3 per group) were infected by PCV2 ($4{\times}10^5$ $TCID_{50}$), PCV2RmA ($2{\times}10^6$ $TCID_{50}$) for 1 week, respectively, and then challenged with $10^5$ $TCID_{50}$ PPV or $10^5$ $TCID_{50}$ PRV for another one week. (L) The serum IFN-β of the infected piglets were measured by ELISA at 24 h post-$2^{nd}$ infection. $^*$ $P < 0.05$, $^{**}$ $P < 0.01$ (compared with PCV2 infected pigs). (M) The viral load of porcine parvovirus (PPV) or porcine pseudorabies virus (PRV) were compared between PCV2 infected pigs and PCV2RmA infected pigs by qPCR. The PPV and PRV loads in the serum of PCV2-infected pigs are higher than that in the serum of PCV2RmA-infected pigs. $^*$ $P < 0.05$, $^{**}$ $P < 0.01$ (compared with PCV2 infected pigs). (N) The histological lesion scores of infection piglets (PCV2 and PCV2RmA).

exhibit significant inhibitory effect on cGAMP and IFN-β production induced by ISD stimulation, PRV or PPV infection, whenever HDAC6 and AKT were silenced/inhibited or not (Fig 8A, 8B and 8C).

As our previous report, gC1qR is a critical regulator of PCV2 nuclear egress [36], thus progeny virion production of PCV2RmA were significantly lower than wild type PCV2 at the same initial infection dose, which led to the viral replication levels in 5 MOI of PCV2RmA-infected cells similar to the viral replication levels in 1 MOI of PCV2-infected cells (Fig 8D). To test whether the reduction of inhibitory effect on IFN-β and cGAMP production in PCV2-infected gC1qR deficient cells or PCV2RmA-infected cells was related to the reduction of viral load in these cells, we used 1 MOI PCV2 and 5 MOI PCV2RmA to compared the induction of cGAMP. Nonetheless, either ISD- or PRV-induced cGAMP was still lower in low-dose PCV2-infected cells than that in high-dose PCVRmA-infected cells (Fig 8E), suggested that the reduction of inhibitory effect in PCV2RmA-infected cells was not due to the reduction of viral load, even though viral reduction indeed weakened the inhibitory effect of PCV2 on IFN-β induction. Next, we conducted a systematic comparison between PCV2RmA mutant and wild-type PCV2 in induction of phosphorylation, poly-ubiquitination, degradation of porcine cGAS. Results showed that the phosphorylation level of porcine cGAS at S278 was lower in PCV2RmA-infected cells than that in PCV2-infected cells (Fig 8F); poly-ubiquitination and degradation of porcine cGAS were also decreased in PCV2RmA-infected cells relative to that in PCV2-infected cells (Fig 8G). Meanwhile, the level of Ac-tubulin (the natural active substrate of HDAC6) increased in PCV2RmA-infected cells compared with PCV2-infected cells (Fig 8H). These results suggested that the interaction of PCV2 Cap with gC1qR is also associated with the activation of HDAC6 during PCV2 infection.

In PCV2RmA infected cells, the levels of PPV and PRV were significantly lower than that in wild-type PCV2-infected cells (Fig 8I), which were further confirmed by standard plaque assay of PRV (S7F Fig). Consistently, culture medium from PCV2RmA-infected cells showed a stronger inhibitory effect on VSV-GFP than the medium from PCV2-infected cells (Fig 8J), since the IFN-β levels was higher in culture medium from PCV2RmA-infected cells (Fig 8K), suggesting that more IFN-β were induced in PCV2RmA-infected cells than wild-type PCV2 infection upon HT-DNA stimulation. Upon PRV or PPV challenge, serum IFN-β were more

significantly upregulated in PCV2RmA-infected pigs than in PCV2-infected pigs (Fig 8L). Likewise, the viral load of PPV and PRV in serum were higher in PCV2-infected pigs than in PCV2RmA-infected pig (Fig 8M), Histopathological examination showed that PRV infection induced lighter histological lesions in the brain and lung of PCV2RmA-infected pigs than that in the PCV2-infected pigs (Figs 8N and S7G). Collectively, these results demonstrated that gC1qR is a critical regulator in PCV2 inhibition of cGAS-mediated innate antiviral responses.

## Discussion

As a central sensor of cytosolic DNA, cGAS mediates the expression of type I interferons and other inflammatory cytokines, while pathogens have evolved diverse mechanisms to suppress cGAS activation, such as masking of cytosolic DNA, dysfunction of cGAS via posttranslational modification, degradation of cGAS by autolysosome or proteasome, or inhibition of the production of second messenger 2′3′-cGAMP [7,40]. In the present study, we found that PCV2 infection phosphorylates cGAS at S278 site via Cap binding protein gC1qR-mediated Akt signaling activation, which directly silences the catalytic activity of cGAS in the early phase of infection. In the later phase of infection, PCV2 appears to induce the formation of autophagic flux, and simultaneously catalyzes the K48-linked poly-ubiquitination of cGAS at K389 and activates the deacetylase activity of histone deacetylase 6 (HDAC6), which recruits K48-ubiquitinated-cGAS through its ubiquitin-binding domain and transports ubiquitinated-cGAS from the cytosol to autolysosome depending on its deacetylase activity, ultimately resulting in cGAS degradation in PCV2-infected cells. Furthermore, PCV2 infection induces the activation of PKCδ, which interacted with HDAC6 and enhanced the deacetylation activity of HDAC6. This finding explains how PCV2 inhibits the type I interferon production to promote self-infection and co-infection of other viruses via targeting cGAS in innate immunity. Importantly, this study reveals PCV2 can inhibit the activation of cGAS signaling pathway through two different mechanisms at different stages of infection, including inhibition of cGAS enzyme activity and degradation of poly-ubiquitinated cGAS, and clarifies the internal relationship and cooperation model between these two mechanisms.

cGAS, also known as MB21D1, is a functionally conserved cytosolic DNA sensor in various species from single eukaryote organisms to mammals [41]. cGAS consists of an N-terminal unstructured domain (residues 1–134 in porcine align to the residues 1–160 in humans) (S5A Fig), followed by a C-terminal nucleotidyltransferase (NTase) domain (residues 135–495 in porcine align to the residues 161–522 in humans) that is required and sufficient for recognizing DNA and producing cGAMP [42–44]. The N-terminal unstructured domain shows a low amino acid sequence homology among different species [45,46]. The C-terminal domain contains a central catalytic pocket and two separate surfaces with positive charges, through which cGAS interacts with the sugar phosphate backbone of the DNA duplex [47–49]. Free cGAS (not bound to DNA) does not show a suitably structured active site. Upon binding to DNA, cGAS assembles into a 2:2 cGAS-dsDNA oligomeric complex and undergoes a global conformational change, which allows substrate ATP and GTP and metal ions to bind in a catalytically proficient manner, leading to the synthesis of 2′3′-cGAMP [44,50]. Notably, on a structural level, DNAs that can activate cGAS contains long dsDNA molecules, single-stranded DNA (ssDNA) with local secondary structure and short, synthetic DNA with G-rich single-stranded overhangs, or RNA-DNA hybrids [51–54]. In this study, we noticed that the ability of cGAS to produce 2′3′-cGAMP was inhibited in PCV2-infected cells whenever stimulated with ISD (a 45-bp non-CpG DNA oligomer which can strongly induce IFN-β) or infected with PRV (a dsDNA virus belongs to the subfamily alpha-herpesvirinae). PCV2 infection promoted the phosphorylation of porcine cGAS at serine 278 through activation of Akt signaling, impaired

its catalytic activity, led to a reduction in cGAMP production. Notably, the serine 278 of porcine cGAS around in a known target motif for Akt kinase that is highly conserved across multiple species, suggesting that regulating Akt activity may be an evolutionarily conservative mechanism used to tightly control the activity of cGAS-STING for avoiding excessive harmful immune response. Interestingly, however, we found that activation of Akt signaling using Akt agonist (SC79) was not able to induce the phosphorylation of porcine cGAS at serine 278. These data suggested that although activation of Akt signaling is necessary for inhibition of cGAS activity, activation of Akt signaling alone is not sufficient for phosphorylation of cGAS to impair its activity. Simultaneously, this result implied that a specific condition must be achieved in PCV2-infected cells for Akt signaling phosphorylation of porcine cGAS and suppressing its enzymatic activity. Currently, it is not known which pathways are involved in this process, although some of signaling pathways have been identified to be activated by PCV2 infection, such as Akt, Erk, p38, JNK, AMPK pathways. Herein, we found that blocking of AMPK signaling can increase cGAS levels through inhibition of autophagic flux induced by PCV2, while blocking of AKT signaling can relieve the suppression of cGAS activity in PCV2-infected cells, suggesting that AMPK signaling and AKT signaling regulate cGAS in different directions during PCV2 infection. Luckily, we further confirm that PCV2 Cap binding protein gC1qR can regulate cGAS in both protein level and activity via regulation of AKT and HDAC6 activation. Firstly, gC1qR is involved in the phosphorylation of cGAS in PCV2-infected cells by the evidences that the phosphorylation level of porcine cGAS at S278 is markedly reduced in either PCV2RmA-infected cells or PCV2-infected gC1qR deficient cells. Despite this change seems reasonable since gC1qR has been identified as a mediator of Akt signaling activation during PCV2 infection in our previous studies [27,36], PCV2 infection was still not able to induce the phosphorylation of porcine cGAS in gC1qR deficient cells in the presence of Akt signaling agonist, suggesting that in the process of PCV2 induction of cGAS phosphorylation, gC1qR not only acts as a mediator to activate Akt pathway, but also participates in the development of a specific condition that may be also required for phosphorylation of porcine cGAS. Secondly, we have identified gC1qR as a key regulator to promote HDAC6 activation in PCV2-infected cells based on the evidences provided in this study that the deacetylation activity of HDAC6 was decreased in either PCV2-infected gC1qR$^{-/-}$ cells or gC1qR-binding activity deficient PCV2 mutant (PCV2RmA)-infected cells and the evidences found in our previous study that the activity of PKCδ was decreased in either PCV2-infected gC1qR$^{-/-}$ cells or PCV2RmA-infected cells[37]. Further deeply study will help us to clearly figure out the regulatory mechanisms of phosphorylation of porcine cGAS, as well as other species of cGAS.

Numerous studies have demonstrated that posttranslational modifications of cGAS and its downstream components are crucial for elaborate controls of the innate immune response to pathogenic DNA. Besides phosphorylation, ubiquitination plays a prominent role in regulating a myriad of important cellular functions [31]. Previous studies have shown that several distinct types of polyubiquitination of cGAS, for example, K27-linked polyubiquitination of human cGAS at Lys173 and Lys384 promotes its enzymatic activity [17]; K48-linked polyubiquitination at Lys271 of human cGAS in uninfected cells and at Lys464 at the late phase of infection promote its proteasomal degradation and contribute to the homeostasis of cGAS for proper initiation and attenuation of an innate immune response [19], and K48-linked polyubiquitination of cGAS at Lys414 by an unknown E3 ligase causes its p62 dependent autophagic degradation at the late phase of viral infection [18]; In PCV2-infected cells, porcine cGAS was also found to be modified by K48-linked polyubiquitination at K389 for p62-dependent autophagic degradation. However, K48-linked polyubiquitination at K389 of porcine cGAS can also be detected in EBSS-treated cell. Intriguingly, K48-polyubiquitinated cGAS was found to be more

easily degraded in PCV2-infected cells than in EBSS-treated cells. Meanwhile, we noted that Histone deacetylase 6 (HDAC6) was activated to contribute the transportation of K48-polyubiquitinated cGAS from cytosol to autolysosome for degradation in PCV2-infected cells, but HDAC6 was not found to be activated and involved in the process of cGAS degradation in EBSS-treated cells. Thus, we conclude that PCV2 infection, but not EBSS treatment, activates HDAC6 to facilitate cGAS transportation, leading to an increased interaction between K48-ubiquitinated cGAS and p62. Based on these findings, we inferred that inactivated HDAC6 may interact with K48-ubiquitinated cGAS but is not able to transport it. Previous study indicates that ubiquitination of cGAS helps to recruit p62 to promote cGAS degradation [18]. In this study, we firstly predicted the potential ubiquitination sites in porcine cGAS through alignment of similar conserved sites between human cGAS and porcine cGAS, and mapped potential ubiquitination sites in porcine cGAS. Difference from the lysine residue at position 414 as a key residue for K48-linked poly-ubiquitination of human cGAS, the lysine residue at position 389 of porcine cGAS is required for its K48-linked poly-ubiquitination whatever PCV2 infection or EBSS treatment, which were evidenced by the data that the K48-linked ubiquitination of K389R cGAS mutant was abrogated in both PCV2-infected cells and EBSS-treated cells. PCV2 infection also promoted the interaction of wild-type cGAS with p62 but failed to promote the interaction of K389R porcine cGAS mutants with p62. Meanwhile, we found that knockdown of HDAC6 reduces the interaction between cGAS and p62 in PCV2-infected cells, but not in EBSS-treated cells, since the transport of K48-ubiquitinated cGAS decrease in PCV2-infected cells due to the decrease of HDAC6. As we all known, HDAC6 acts as the only cytoplasmic deacetylase to regulate diverse cellular processes including autophagy, the ubiquitin-proteasome system, and cell migration by enzyme-dependent and -independent mechanisms [55]. Our results suggested that HDAC6 may be a crucial player in mediating the degradation of ubiquitinated cGAS during diverse viruses' infection, which needs to be explored in future. Meanwhile, this study showed that PCV2 Cap act as a predominant regulator to promote porcine cGAS phosphorylation and HDAC6 activation through mediating PI3K/AKT and PKCδ signaling activation. Importantly, PCV2 Cap-binding protein gC1qR (also known as C1QBP/gC1qR/HABP1/p32), a multiligand-binding and multifunctional protein [56,57], is also a crucial regulator for the activation of HDAC6 and HDAC6/autophagy-mediated cGAS degradation in PCV2-infected cells. Taken together, these data demonstrated that PCV2 infection activates the gC1qR/HDAC6/autophagy regulator axis to promote cGAS degradation.

The cGAS-STING pathway is precisely regulated in spatial and temporal dimensions to efficiently control the innate immune response for elimination of pathogens and termination of inflammatory response timely to avoid inflammatory injury. Previous studies have provided numerous evidences that various post-translational modifications on cGAS regulate cGAS protein level and activity to control the activation of cGAS-STING pathways. However, how these posttranslational regulations are coordinated to dynamically regulate its enzyme activity and degradation remain open to interpretation. In this study, we have shown that the phosphorylation of porcine cGAS at S278 facilitates the K48-linked poly-ubiquitination and degradation of porcine cGAS in PCV2-infected cells. This is a new discovery in the post-translational modification of cGAS between phosphorylation and ubiquitination and paints a picture for the dynamic regulation of cGAS protein level and activity in delineating the mechanism of porcine circoviruses inhibition of type I interferon induction.

As the protagonist of the story, circoviruses are nonenveloped DNA viruses with a single-stranded DNA genome [58]. They can infect various domestic and wildlife animal species such as mammals, fish, avian species, and even insects [59]. PCV2, as the most prevalent pathogen in global pig populations [60], leads to the infected pigs more susceptible to other

pathogens [39]. Our previous studies have showed that PCV2 infection suppresses IL-12p40 expression to lower host Th1 immunity through the Cap and gC1qR interaction–mediated PI3K/Akt and p38 MAPK pathway and employs IL-10 to block the transfer of T cells, promotes the polarization of macrophage M2, and attenuates the production of pro-inflammatory cytokines [27,61]. These studies provide a new theory for explaining why PCV2 infection increases susceptibility to other pathogens. As a robust first line of host innate defense against pathogenic infection, type I interferon system should be addressed or employed as a kind of escape innate immunity strategy by circoviruses. Indeed, as we guessed that PCV2 did inhibit type I interferon expression via suppressing cGAMP production by gC1qR-mediated catalytic activity inhibition in the early phase of infection, and gC1qR/HDAC6/autophagy regulator axis-mediated cGAS degradation in the late phase of infection. During this process, gC1qR not only participates in suppressing the catalytic activity of cGAS through activation of Akt signaling or other undetermined signalings, but also recruited PKCδ to promote the activation of HDAC6, confirming that gC1qR is a crucial regulator during PCV2 suppression of anti-virus innate immunity. On the basis of this finding and our previous studies, we confirmed that the gC1qR-binding activity deficient PCV2 mutant (PCV2RmA) not only did not induce obvious lesions in infected pigs, but also did not increase the susceptibility of infected pigs to other pathogens. Thus, these findings provide beneficial evidences for development of this PCV2 mutant into an attenuated vaccine.

In summary, the data presented in this work demonstrate that PCV2 infection promotes the infection various DNA viruses through targeting porcine cGAS to inhibit IFN-β induction relying on followed mechanisms (Fig 9). In the early phase of PCV2-infection, porcine cGAS is phosphorylated at S278 site via Cap binding protein gC1qR-mediated PI3K/AKT signaling activation, which directly silences the catalytic activity of cGAS to inhibit cGAMP production; Subsequently, phosphorylation of cGAS at S278 facilitates the K48-linked poly-ubiquitination of cGAS at K389, while PKCδ is phosphorylated and recruited by gC1qR to promote HDAC6 activation; then K48-ubiquitinated-cGAS proteins are recruited and transported from the cytosol to autolysosome by activated histone deacetylase 6 (HDAC6) depending on its deacetylase activity and ubiquitin-binding function, thereby eventually resulting in a markedly increased cGAS degradation in PCV2 infection-induced autophagic cells. On the basis of the roles of gC1qR in PCV2 suppression of anti-virus innate immunity, we have provided strong evidence for developing gC1qR-binding activity deficient PCV2 mutant (PCV2RmA) as a better vaccine candidate strain. Taken together, our findings uncover a systemic mechanism by which PCV2 infection inhibits type I interferon induction, which helps us further develop a better vaccine for control of PCV2.

## Materials and methods

### Ethics statement

All animal experiments were approved by the Institutional Animal Care and Use Committee (IACUC) of Northwest A&F University (permit numbers: 20190716 and 20200619) and were performed according to the Animal Ethics Procedures and Guidelines of the People's Republic of China. No other specific permissions were required for these activities. This study did not involve endangered or protected species.

### Cell culture and virus preparation

HEK 293T cells were purchased from the American Type Culture Collection (ATCC); porcine PK-15 (kidney 15 cell line) cells were stocks in our lab [36]. These cells were cultured in Dulbecco's modified Eagle's medium (DMEM) supplemented with 10% fetal bovine serum and

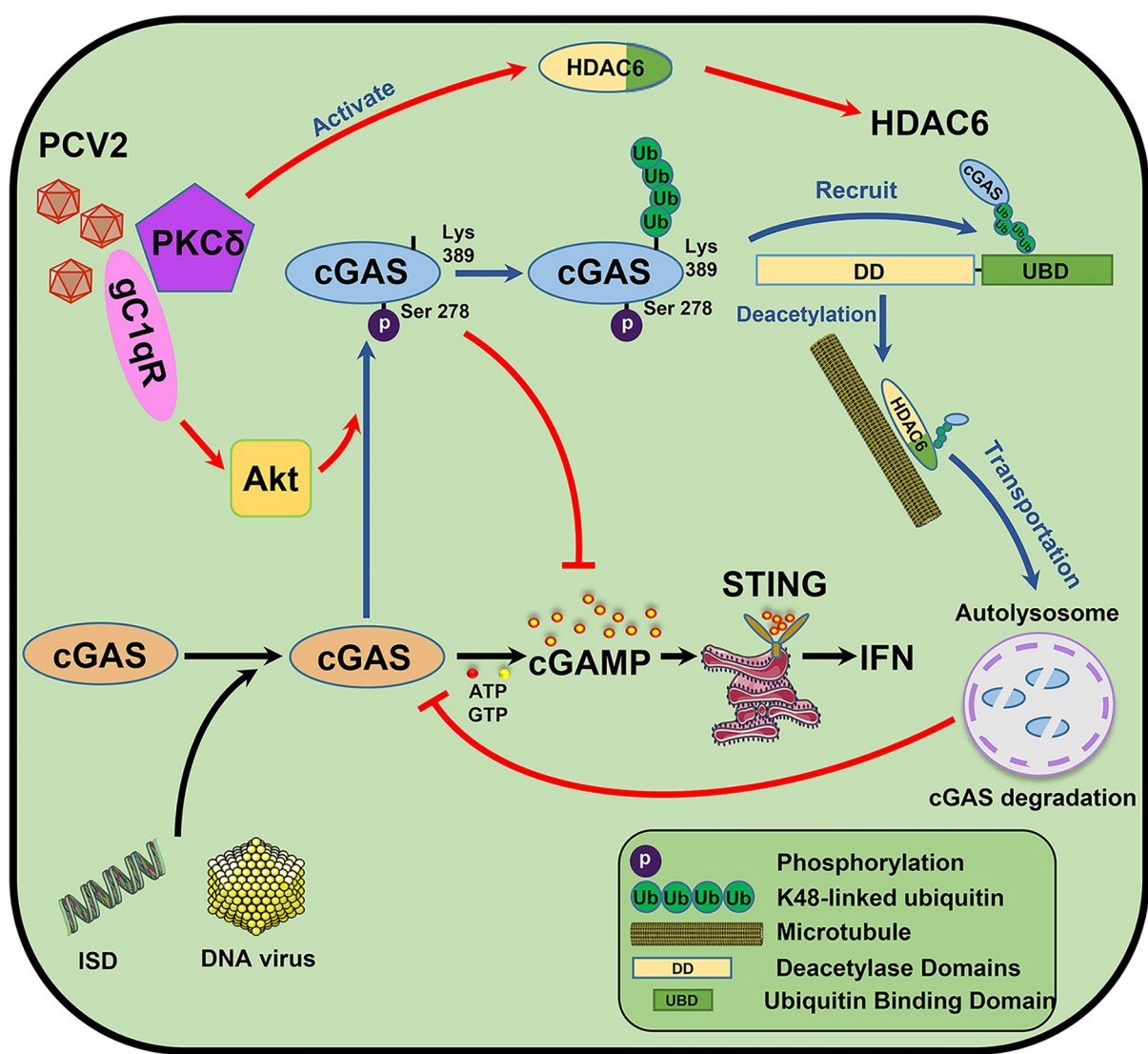

**Fig 9. Model of PCV2 targeting cGAS to suppress Type I interferon production by gC1qR-mediated catalytic activity inhibition and HDAC6-mediated autophagic degradation.** In the early phase of PCV2-infection, porcine cGAS is phosphorylated at S278 site via Cap binding protein gC1qR-mediated PI3K/AKT signaling activation, which directly silences the catalytic activity of cGAS to inhibit cGAMP production; Subsequently, phosphorylation of cGAS at S278 facilitates the K48-linked poly-ubiquitination of cGAS at K389, while PKCδ is phosphorylated and recruited by gC1qR to promote HDAC6 activation; then K48-ubiquitinated-cGAS proteins are recruited and transported from the cytosol to autolysosome by activated histone deacetylase 6 (HDAC6) depending on its deacetylase activity and ubiquitin-binding function, thereby eventually resulting in a markedly increased cGAS degradation in PCV2 infection-induced autophagic cells.

penicillin/streptomycin (100 U/mL) and were maintained at 37°C in a humidified atmosphere with 5% $CO_2$.

Wild-type PCV2 strain (genotype PCV2b; GenBank accession number MH492006.1), PPV (Genbank: MK993540), PRV (Genbank: MH582511.1) and the p32 binding site mutant PCV2 strain (PCV2RmA) are isolated or constructed in our previous study [36,62].

## Plasmids and reagents

Porcine cGAS and the cGAS mutants [cGAS(S278A), cGAS(S278D), and cGAS(K389R)] were amplified by overlap PCR, with a 3×Flag tag sequence, all sequences were confirmed by

sequencing analysis (Sangon Biotech). Ubiquitination plasmids were purchased from Addgene (22900, 22901, 22902, 22903, 17607, 17604, 17605, 17606). Porcine IFN-β (Interferon Beta) ELISA Kit (ES8RB) were purchased from Thermo Fisher Scientific. Autophagy flux inhibitor Bafilomycin A1 (54645) was purchased from Cell Signaling Technology. Autophagy inhibitor 3MA (5142-23-4) was purchased from Sigma-Aldrich (USA). Protein G-agarose (sc-2002) and protein A-agarose (sc-2001) were purchased from Santa Cruz. DAPI (C1005) was purchased from Beyotime Biotechnology. PI3K/AKT inhibitor (LY294002), ERK1/2 MAPK inhibitor (PD98059), and p38 MAPK inhibitor (SB203580) were purchased from Merck; JNK inhibitor (SP600125) and AMPK inhibitor (Compound C•2HCl), cGAMP sodium salt (SML1229), Protease Inhibitor Cocktail (P8340) were purchased from Sigma-Aldrich. Tubastatin A (Hydrochloride) (HY-13271) and Akt agonist (HY-18749) were purchased from MCE. Rabbit anti-IRF3 (4302) monoclonal antibody, rabbit anti-phospho-IRF3 monoclonal antibody (4947), mouse anti-TBK1/NAK monoclonal antibody (51872), rabbit anti-phospho-TBK1/NAK monoclonal antibody (5483), mouse anti-ubiquitin monoclonal antibody (3936), rabbit anti-acetyl-α-tubulin monoclonal antibody (5335), rabbit anti-HDAC6 monoclonal antibody (7612), rabbit anti-Phospho-Akt Substrate monoclonal antibody (9614) that recognizes peptides and proteins containing phospho-(Ser/Thr) Akt substrate motif (RXXS*/T*), rabbit anti-histone H3 monoclonal antibody (4499), rabbit monoclonal anti-PKC-δ antibody (9616) were purchased from Cell Signaling Technology. Rabbit anti-SQSTM1/p62 polyclonal antibody (ab101266), mouse anti-HDAC6 monoclonal antibody (ab253032), goat anti-HA polyclonal antibody (ab215069), Goat polyclonal Secondary Antibody to Rabbit IgG-H&L (Alexa Fluor 647) (ab150079) were purchased from Abcam PLC (Abcam, Cambridge, UK). Mouse anti-HA monoclonal antibody (H3663), Mouse anti-Myc monoclonal antibody (M5546) were purchased from Sigma Aldrich. Rabbit anti-Flag monoclonal antibody (701629), mouse anti-Flag monoclonal antibody (MA1-91878), horseradish peroxidase (HRP)conjugated anti-mouse IgG (31430), and anti-rabbit IgG (31460), Rabbit anti-Goat IgG (H+L) Cross-Adsorbed Secondary Antibody, Alexa Fluor 647 (A-21446) were purchased from Thermo Fisher. Anti-β-actin (A00702) and anti-α-tubulin (A01410-100) were obtained from GenScript Biotech Corporation. Fluorescein isothiocyanate (FITC)-conjugated goat anti-mouse IgG antibody (BA1101), DyLight 594 Conjugated goat Anti-Mouse IgG (BA1141), DyLight 594 Conjugated Goat Anti-Rabbit IgG (BA1142) were purchased from Boster. The anti-cGAS polyclonal antibody was obtained from a mouse immunized with purified full-length cGAS protein expressed by the pET32a vector in *Escherichia coli* (*E. coli*).

## Animal experiment

Five-week-old cross-bred piglets were purchased from a native herd free of PCV2, PRV, porcine parvovirus, and other major swine pathogens as determined by PCR. All piglets were housed under the same conditions and treated similarly. For the first experiment presented in Fig 1E and 1F, piglets were randomly divided into six groups (n = 3, per group), and inoculated with PCV1 ($4×10^5$ TCID$_{50}$), PCV2 ($4×10^5$ TCID$_{50}$), or mock (same volume of medium) for 1 week, respectively, and then challenged with $10^5$ TCID$_{50}$ PPV or $10^5$ TCID$_{50}$ PRV for another 24 h. and then the serum IFN-β of the infected piglets were measured by ELISA; IFN-β, IFIT1, and CXCL10 mRNA levels in lung tissues were determined by qPCR. For the second experiment presented in Figs 8L, 8M, 8N and S7G, piglets were divided into four groups (n = 3, per group) to inoculate PCV2 ($4×10^5$ TCID$_{50}$), PCV2RmA ($2×10^6$ TCID$_{50}$) for 1 week, respectively, and then challenged with $10^5$ TCID$_{50}$ PPV or $10^5$ TCID$_{50}$ PRV for another 1 week. The viral copy numbers of PPV and PRV in the serum of PCV2-infeced pigs and

PCV2RmA-infected pigs were measured by qPCR, and immunohistochemistry (IHC) staining for PRV and hematoxylin and eosin (H&E) staining in the brain and lung were observed.

## siRNA Transfection

Specific siRNAs used to silence Akt, p38 MAPK, JNK, ERK, AMPK, PKCδ were referred to in our previous study[36,63]. HDAC6 (GenBank: XM_003360315.5) and AMPK (GenBank: NM_001167633.1) siRNAs were designed (HDAC6 siRNA: 5′-GGAGGAGCUUAU-GUUGGTT-3′; AMPK siRNA: 5′-GCUUGCCAAAGGAAUGATT-3′). Specific siRNAs or negative control (NC) siRNA were transfected into PK-15 cells using Lipofectamine 3000 (Invitrogen) according to the manufacturer's instructions. The effects of siRNAs were identified by Western blotting.

## RNA extraction and quantitative RT-PCR

The RNA extraction was isolated using TRIzol reagent (Invitrogen) according to the manufacturer's instructions. Equal amounts of total RNA (2 μg) were used to synthesize cDNA using the PrimeScript RT Master Mix (TaKaRa). Gene expressions were analyzed by quantitative reverse transcription-PCR (qPCR) with Applied Biosystems QuantStudio 6&7 (Applied Biosystems, Grand Island, NY, USA) using SYBR Premix Ex Taq II DNA polymerase (TaKaRa). Specific primers used for IFN-β, IFIT1, and CXCL10 were used for real-time PCR: Sus scrofa IFN-β forward primer, 5′- AACCACC ACAATTCCAGAGGG -3′, reverse primer 5′- GGTTT CATTCCAGCCAGTGC -3′, Sus scrofa IFIT1 forward primer, 5′- TCAGAGGTGAGAAGG CTGGT -3′, reverse primer 5′- GCTTCCTGCAAGTGTCCTTC-3′, Sus scrofa CXCL10 forward primer, 5′-TTCGCTGTACCTGCATCAAG-3′, reverse primer 5′-CAACATGTGGG CAAGA TTGA-3′. The data were analyzed by The QuantStudio 6 and 7 Flex Real-Time PCR System Software.

## CRISPR-Cas9-mediated porcine cGAS knockout in PK-15 cells

CRISPR-Cas9 design and analysis method were described in a previous study [64]. Three pairs of oligos were designed based on the porcine cGAS sequence (XM_013985148.2). Oligonucleotides were annealed and ligated into the Lenti-CRISPRv2 plasmid (catalog number 52961; Addgene) by use of the *Bsm*B I site, respectively, and confirmed by sequencing analysis (Sangon Biotech). And the recombinant plasmids were co-transfected with recombinant plasmids accompanied with psPAX2 (catalog number 12260; Addgene) and pMD2.G (catalog number 12259; Addgene) into 293T cells to obtain recombinant lentivirus. The recombinant lentiviruses were used to infect PK-15 cells, respectively. 5 μg/mL puromycin (InvivoGen) was added into the cell cultures to select the knockout cells 48 h later. The selected positive cells were obtained after about 10 days and then subcloned into 96-well plates for single-clone growth and saved as cell stocks, finally, the positive cells were checked by western blotting and sequencing.

## Coimmunoprecipitation and Western blotting

The cells cultured in 100-mm-diameter dishes (Thermo Fisher) were transfected with indicated plasmids using Lipofectamine 3000, and 36 h later, the cells were lysed with lysis buffer (150 mM NaCl, 50 mM Tris-HCl [pH 7.4], 1% Nonidet P-40, 0.5% Triton X-100, 1 mM EDTA, 0.1% sodium deoxycholate, 1 mM dithiothreitol, 0.2 mM phenylmethylsulfonyl fluoride, and a protease inhibitor cocktail [Sigma-Aldrich]) on ice for 30 min. The cell lysate supernatant was collected by centrifuging and then precleared by incubation with protein G/

protein A agarose for 1 h at 4˚C. The supernatant was incubated with indicated antibodies overnight at 4˚C and precipitated with protein G agarose/protein A agarose for 30 min at room temperature. Then, the precipitated protein G agarose/protein A agarose was centrifuged at $2,000 \times g$ for 10 s and washed three times with PBS. Finally, the bound proteins were eluted by boiling for 10 min in $2 \times$ loading buffer, followed by SDS-PAGE and immunoblotting. Immunoreactive bands were visualized using enhanced chemiluminescence (ECL) reagents (Bio-Rad).

## Confocal microscopy

Cells grown on coverslips in 24-well culture plates were transfected with indicated plasmids or infected with the virus for the indicated time. Then the cells were fixed with 4% paraformaldehyde for 20 min at room temperature and permeabilized with 0.1% Triton X-100 for 15 min at room temperature. After washing with 0.1 M phosphate-buffered saline (PBS) and preincubating with 2% bovine serum albumin for 1 h at 37˚C, the cells were successively incubated with primary antibodies overnight and with secondary antibodies for 1 h at 37˚C, followed by thrice washing as described above. After quick staining with DAPI, coverslips were mounted onto glass slides in the presence of a fluorescence mounting medium. Samples were analyzed by Leica TCS SP8 laser scanning confocal microscope. Images were recorded using Leica X software.

## cGAMP activity assay

cGAMP activity assay was performed as previously described [65,66]. Infected or transfected PK-15 cells were washed with phosphate buffer saline (PBS) and lysed in hypotonic buffer (10 nM Tris, pH 7.4, 10 mM KCl, 1.5 mM $MgCl_2$). The cell extracts were incubated with 1 kU/mL Benzonase for 30 min at 37˚C, then heated at 95˚C for 5 min and centrifuged for 5 min at maximum speed ($16,000 \times g$). PK-15 cells were permeabilized as described previously [67], with modifications. Briefly, medium was aspirated from the PK-15 cells, and digitonin permeabilization solution (50 mM HEPES pH 7.0, 100 mM KCl, 85 mM sucrose, 3 mM $MgCl_2$, 0.2% BSA, 1 mM ATP, 0.1 mM DTT, 10 μg/mL digitonin) was added to treated the cell extracts. PK-15 reporter cells were incubated with extracts for 30 min at 37˚C and then replace with a supplemented medium. RNA was harvested at 6 h after the initial addition of extracts and qPCR for IFN-β, as a marker of cGAMP activity.

## LC-MS of in vivo cGAMP synthesis assay

According to a previously published protocol [68], unmodified cGAS protein and serine phosphorylation cGAS protein were incubated with HT-DNA (20 μg/mL) in reaction buffer (20 mM HEPES, PH 7.5, 5 mM $MgCl_2$, 2 mM ATP, 2 mM GTP) at 37˚C for 2 h. The samples were diluted by 5-fold and centrifuged at $16,000 \times g$ for 10 min, and the supernatant was filtered with a 10 kD ultrafiltration filter (Millipore). Following vacuum dry, the cGAMP was reconstituted in 300 μL of 50% acetonitrile and was ultrasonicated for 30 min at room temperature. After centrifugation at $14,000 \times g$ for 15 min, the supernatant was collected for LC-MS/MS analysis. The LC-MS/MS analysis was performed on an LC-30A (Shimadzu, JPN) coupled to an AB Sciex Triple TOF 5600[+] mass spectrometer (AB Sciex, USA) with the electrospray ionization (ESI) source. A Waters ACQUITY UPLC BEH Amide column (1.7 mm, $2.1 \times 100$ mm) was used for GAMP separation with a flow rate at 0.3 mL/min and column temperature of 45˚C. The mobile phases were comprised of (A) 0.2% formic acid and 10 mM ammonium acetate in 50% acetonitrile and (B) 0.2% formic acid and 10 mM ammonium acetate in 95% acetonitrile. The gradient elution was 80% B kept for 1.0 min, then changed linearly to 5% B during

7.0 min, increased to 80% B in 7.1 min and maintained for 2.9 min. The injection volume was set to 10 μL. The mass parameters were as follows: ion spray voltage was 5500 V, ion source temperature was 550˚C, collision gas was Medium, ion source gas 1 was 50 psi, ion source gas 2 was 50 psi, curtain gas was 35 psi.

## Statistical analysis

Data are represented as mean ± standard error of the mean (SEM) (standard deviation [SD]). Statistical analyses were performed with GraphPad Prism 8 software using two-way ANOVA followed by Bonferroni post hoc test or unpaired Student $t$-test. Immunofluorescence values were calculated using Image-Pro Plus 6.0. Statistically significant and very significant results were defined as $P < 0.05$ and $P < 0.01$.

## Supporting information

**S1 Fig. PCV2 infection inhibits other DNA virus-induced type I interferon production and response. (A, B)** PCV2 inhibits PPV- or PRV-induced IFN-β production and response. The PK-15 cells were infected by PCV1 (MOI = 5), PCV2 (MOI = 5), or Mock (same volume of medium) for 48 h, respectively, and then challenged with 1 MOI PPV, PRV or ISD for another 6 h. The supernatant IFN-β levels were measured by ELISA (A). IFN-β, IFIT1, and CXCL10 mRNA levels were determined by Q-PCR (B). $^{*}$ $P < 0.05$, $^{**}$ $P < 0.01$ (compared with mock infection); $^{\#}$ $P < 0.05$, $^{\#\#}$ $P < 0.01$ (compared with PCV1 infection).
(TIF)

**S2 Fig. PCV2 inhibits the activation of type I IFN signaling to facilitates the infection of other DNA viruses by reducing cGAMP production. (A-D)** The wild type PK-15 cells (A, B) and IFNAR$^{-/-}$ PK-15 cells (C, D) were infected with PCV2 (MOI = 5) for the indicated time (A, C), or infected with different doses (0.1, 1, 10 MOI) of PCV2 for 48 h (B, D), and then the relative IFN-β mRNA levels were determined by Q-PCR at 6 h following ISD stimulation or PPV or PRV infection. **(E)** PK-15 cells were infected with PCV2 (MOI = 5) for the indicated time, and then the relative cGAMP levels were determined at 6 h following ISD stimulation or PRV infection. $^{*}$ $P < 0.05$, $^{**}$ $P < 0.01$, compared with infection at 0 h (A, C), 0 MOI PCV2 (B, D), or Mock infection (E).
(TIF)

**S3 Fig. EBSS treatment promotes the K48-linked poly-ubiquitination of porcine cGAS at K389 for subsequent degradation. (A)** EBSS treatment induces the poly-ubiquitination of cGAS. PK-15 cells were treated with EBSS along with or without Baf or CQ for 48 h. Cell lysates were analyzed by immunoprecipitated with anti-porcine cGAS antibody, and ubiquiti-nated cGAS proteins were immunoblotted using anti-ubiquitin antibodies. **(B)** Alignment of cGAS amino acid partly sequences. Highlighted amino acids indicate conserved lysine (K) of cGAS. **(C)** EBSS treatment promotes the K48-linked poly-ubiquitination of porcine cGAS at K389. PK-15 cells were transfected with different HA-Ub constructs as indicated, then were treated with EBSS along with Baf for 48 h. Cell lysates were immunoprecipitated with anti-Flag antibody and immunoblotted with anti-HA antibody. **(D)** K389R cGAS mutant degradation was abrogated in EBSS-treated cells. PK-15 cells were transfected with plasmids as indicated, then were treated with EBSS for the indicated time. cGAS levels were analyzed by western blot-ting. **(E)** Poly-ubiquitination of porcine cGAS at K389 is required for the interaction of cGAS with p62. The cGAS$^{-/-}$ PK-15 cells expressed Flag-cGAS, Flag-cGAS (K389R) were infected with PCV2 in the presence of Baf. The localization of porcine cGAS and PCV2 Cap protein was observed under confocal microscopy. Scale bar, 10 μm. **(F)** Co-immunoprecipitation

experiment to test the affinity of WT and K389R cGAS to p62 in the cGAS$^{-/-}$ PK-15 cells that transfected with HA-p62 and 3×Flag-cGAS or 3×Flag-cGAS (K389R) constructs.
(TIF)

**S4 Fig. PCV2 infection activates HDAC6 and promotes the interaction of ubiquitinated cGAS with p62 via HDAC6 mediation. (A)** Detection of the acetylated tubulin levels to determine the deacetylase activity of HDAC6 in PCV2 or mock infection cells. Statistical analysis of the Ac-tubulin levels in the indicated samples. Scale bar, 10 μm. $^{**}$ $P < 0.01$. **(B)** PK-15 cells were transfected with HDAC6 specific siRNA (siHDAC6) or siRNA negative control (siN.C.) and other plasmids as indicated for 24 h. Then infected with PCV2 (MOI = 5) or mock in the presence of Baf for 48 h. The poly-ubiquitination levels and protein levels of cGAS were analyzed. **(C)** PK-15 cells were transfected with HDAC6 specific siRNA (siHDAC6) or siN.C. and other plasmids as indicated for 24 h. Then infected with PCV2 (MOI = 5) or mock in the absence of Baf for 48 h. The poly-ubiquitination levels and protein levels of cGAS were analyzed. **(D)** PK-15 cells were transfected with (siHDAC6) or siN.C. for 24 h, then treated with EBSS to detect the levels of porcine cGAS, HDAC6, and Ac-Tubulin at indicated times. **(E)** Detection of the acetylated tubulin levels to determine the deacetylase activity of HDAC6 in EBSS-treated or untreated cells. **(F)** The cGAS$^{-/-}$ PK-15 cells transfected with Flag-cGAS, Flag-cGAS (K389R) expression constructs were infected with PCV2 in the presence of Baf. The localization of porcine cGAS and PCV2 Cap protein was observed under confocal microscopy. Scale bar, 10 μm. **(G)** PK-15 cells were infected with PCV2 in the presence of Baf, then the colocalization of porcine cGAS, HDAC6, K48-Ub, and p62 were observed under confocal microscopy. Scale bar, 10 μm. **(H)** PK-15 cells transfected indicated plasmids were treated with Tub A for 6 h, then infected with PCV2 (MOI = 5) for another 48 h, and the interaction of ubiquitinated cGAS with p62 was analyzed. **(I)** PK-15 cells were pretreated with Tub A and infected with PCV2 (MOI = 5) for the indicated time, and then the levels of porcine cGAS, PCV2 capsid, and Ac-Tubulin were determined by western blotting. **(J)** PK-15 cells expressed Flag-cGAS were treated with EBSS for 48 h, and then the interaction of ubiquitinated cGAS with p62 was analyzed. **(K)** PK-15 cells were pretreated with Tub A and treated with EBSS for the indicated time, and then the levels of porcine cGAS, HDAC6, and Ac-Tubulin were determined by western blotting.
(TIF)

**S5 Fig. PCV2 induces phosphorylation of porcine cGAS at Ser278 to negatively regulate cGAS enzymatic activity. (A)** Alignment of cGAS sequences between human and pig. **(B)** The purification of porcine cGAS WT, cGAS S278A, and cGAS S278D protein for in vitro enzymatic assay. Visualized by Coomassie brilliant blue staining. **(C)** LC-MS analysis of cGAMP production from an in vitro cGAMP synthesis assay. Small molecules were extracted from in vitro tubes for analysis of cGAMP isomers by tandem mass spectrometry. **(D)** The cGAS$^{-/-}$ PK-15 cells reconstituted with the WT cGAS, or cGAS mutant S278A, or cGAS mutant S278D were infected with mock or PCV2 in the presence of Baf for 12 h, and then these protein levels were detected by western blotting.
(TIF)

**S6 Fig. The phosphorylation of porcine cGAS at Ser278 facilitates the K48-linked poly-ubiquitination and degradation of porcine cGAS during PCV2 infection. (A-B)** PK-15 cells transfected with indicated siRNA were infected with PCV2 (MOI = 5) for 12 h, and then the relative cGAMP production levels were determined by report assay at 6 h following ISD stimulation or PRV infection (A); the phosphorylation level of cGAS at the S278 site was detected by western blotting (B). $^{*}$ $P < 0.05$, $^{**}$ $P < 0.01$ (compared with siN.C.). **(C)** PK-15 cells were

treated with Akt agonist (SC79) for indicated times, and the phosphorylation level of cGAS at the S278 site was detected by western blotting. **(D-E)** The phosphorylation of cGAS facilitates the ubiquitination and degradation of cGAS during PCV2 infection. PK-15 cells were transfected with indicated expression constructs and indicated siRNA, and then infected with PCV2 (MOI = 5) in the presence or absence of Baf to detect the poly-ubiquitination levels and protein levels of cGAS. **(F)** EBSS-induced cGAS ubiquitination is independent of the phosphorylation. The cGAS$^{-/-}$ PK-15 cells were reconstituted with the WT cGAS, S278A mutant, or S278D mutant, treated with EBSS to detect the poly-ubiquitination levels and phosphorylation levels of cGAS. **(G)** The wild-type PK-15 cells and cGAS$^{-/-}$ PK-15 cells were transfected with Flag-cGAS or Flag-cGAS (S278A) expression constructs, then infected with PCV2 to observe the localization of porcine cGAS and PCV2 Cap protein. Scale bar, 10 μm.
(TIF)

**S7 Fig. gC1qR is a critical negative regulator in PCV2 promoting other DNA virus infection. (A, B)** gC1qR deficiency diminishes the inhibitory effects of PCV2 on cGAMP and IFN-β induction. gC1qR$^{-/-}$ PK-15 cells and wild type PK-15 cells were infected with wild type PCV2 (MOI = 5) for the indicated time, and then the relative cGAMP production levels and IFN-β mRNA levels were determined at 6 h following ISD stimulation or PPV/PRV infection. $^{*}$ $P < 0.05$, $^{**}$ $P < 0.01$ (compared with wild-type cells). **(C, D)** PCV2 Cap inhibits the induction of cGAMP and IFN-β. PK-15 cells pretreated with Baf were infected with rAd-Blank (MOI = 100), rAd-Rep (MOI = 100) and rAd-Cap (MOI = 100) for 24 h, and then the relative cGAMP production levels and IFN-β mRNA levels were determined at 6 h following PPV or PRV infection by report assay (C) and qPCR respectively (D). $^{**}$ $P < 0.01$ (compared with rAd-Blank). **(E)** gC1qR$^{-/-}$ PK-15 cells and wild type PK-15 cells were treated with Akt agonist (SC79) for 6h, then infected with Mock or PCV2 along with Baf. **(F)** PCV2RmA is a weak strain relative to PCV2 in promotion of DNA virus replication. PK-15 cells were infected with PCV2 (MOI = 1) or PCV2RmA (MOI = 5) for 48 h, then were further infected with PRV, and the relative viral titers were measured by standard plaque assay. Viral plaques were observed and the viral titer was calculated. $^{*}$ $P < 0.05$, $^{**}$ $P < 0.01$. **(G)** PCV2RmA alleviating PRV-induced pathological changes. The piglets were infected by PCV2 ($4 \times 10^5$ TCID$_{50}$), PCV2RmA ($2 \times 10^6$ TCID$_{50}$) for 1 week, respectively, and then challenged with $10^5$ TCID$_{50}$ PRV for another week. Representative images of immunohistochemistry (IHC) staining for PRV using Mouse anti-gC polyclonal antibody (lower panel) and hematoxylin and eosin (H&E) staining (upper panel) in the brain and lung derived from PCV2-, PCV2RmA-, or PBS (Mock) groups challenged with PRV. Bar, 100 μm.
(TIF)

## Acknowledgments

We would like to thank Prof Shan-Lu Liu (The Ohio State University, Columbus, Ohio, USA) for correction and revision of this manuscript in English. We also thank the Life Science Research Core Services (LSRCS) NWAFU (Yanqing Wang) for assistance with immunofluorescence and (Luqi Li, Junmin Li, and Meijuan Ren) for the technical support.

## Author Contributions

**Conceptualization:** Zhenyu Wang, Dewen Tong, Yong Huang.

**Data curation:** Zhenyu Wang, Jing Chen.

**Formal analysis:** Zhenyu Wang, Jing Chen, Xingchen Wu, Qian Du.

**Funding acquisition:** Dewen Tong, Yong Huang.

**Investigation:** Zhenyu Wang, Dan Ma, Xiaohua Zhang, Cong Han, Haixin Liu, Xiangrui Yin, Qian Du.

**Methodology:** Zhenyu Wang, Jing Chen, Xingchen Wu, Dan Ma, Xiaohua Zhang, Ruizhen Li, Cong Han, Haixin Liu, Qian Du.

**Project administration:** Zhenyu Wang, Jing Chen, Xingchen Wu, Dan Ma, Xiaohua Zhang, Ruizhen Li, Qian Du, Dewen Tong, Yong Huang.

**Supervision:** Dewen Tong, Yong Huang.

**Validation:** Zhenyu Wang, Jing Chen, Xingchen Wu, Yong Huang.

**Writing – original draft:** Zhenyu Wang, Yong Huang.

**Writing – review & editing:** Zhenyu Wang, Jing Chen, Dewen Tong, Yong Huang.

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
