## [Decision Letter · Decision Letter 0]

24 May 2021

Dear Dr. Huang,

Thank you very much for submitting your manuscript "PCV2 targets cGAS to inhibit type I interferon induction to promote other DNA virus infection" for consideration at PLOS Pathogens. As with all papers reviewed by the journal, your manuscript was reviewed by members of the editorial board and by several independent reviewers. In light of the reviews (below this email), we would like to invite the resubmission of a significantly-revised version that takes into account the reviewers' comments.

We cannot make any decision about publication until we have seen the revised manuscript and your response to the reviewers' comments. Your revised manuscript is also likely to be sent to reviewers for further evaluation.

Sincerely,

Pinghui Feng

Associate Editor

PLOS Pathogens

Blossom Damania

Section Editor

PLOS Pathogens

Kasturi Haldar

Editor-in-Chief

PLOS Pathogens

orcid.org/0000-0001-5065-158X

Michael Malim

Editor-in-Chief

PLOS Pathogens

orcid.org/0000-0002-7699-2064

Reviewer's Responses to Questions

**Part I - Summary**

Reviewer #1: This is a complete and complex study, investigating the molecular mechanisms by which PCV2 impairs cGAS signaling and interferon production. The story began with PCV2 infection promotes the infection of other DNA viruses. One mechanism was identified as inhibiting cGAS-mediated IFN production, which was validated with ISD stimulation. The authors then went further to examine which pathways or cellular factors are involved, and successfully discovered that PCV2 causes cGAS K48-linked polyubiquitination at K398, and subsequent degradation in autolysosomes, with activation of HDAC6. One layer of complexity is PCV2, through viral Cap proteins an host factor gC1pR, causes cGAS phosphorylation at S278 which strongly suppresses cGAS enzymatic activity to synthesize cGAMP, the secondary messenger to activate STING. This second mechanism explains why when PCV2-triggered degradation of cGAS is rescued with autophagy inhibitors, IFN production is still impaired. Collectively, this study convincingly demonstrates that PCV2 uses its Cap protein to engage gC1pR, activates Akt and HDAC6, causes cGAS phosphorylation and polyubiquitination, which leads to impairment of cGAS function and IFN response.

Reviewer #2: Wang et al. reported a PCV2 viral immune evasion strategy targeting IFN signaling, by phosphorylating cGAS to inhibit its catalytic ability and degrading cGAS by K48-Polyubiquitination-mediated and HDAC6-facilitated autophagy. Overall speaking, the completeness of the study and the quality of the data meet the standard of Plos Pathogen. However, I have major concerns over the novelty of the manuscript as cGAS phosphorylation and K48-Ub-mediated autophagic degradation have been published five years ago. Taken account of these published findings, the novelty of paper is limited to: (1) Cap-gC1qR promotes PKCδ to activate Akt; (2) Cap-gC1qR promotes HDAC6 to facilitate cGAS degradation; (3) a PCV2 mutant (PCV2RmA) serving as a potential vaccine.

Reviewer #3: In this manuscript, Wang et al. studied how PCV2 restricts cGAS-mediated innate immune signaling. Authors found that PCV2 infection induced K48-linked ubiquitination of cGAS at K389, which led to the autophagosomal degradation of cGAS depending on HDAC6. Then the authors further showed that cGAS was phosphorylated at S378 by AKT, which seems to precede the ubiquitination of K389. Moreover, authors found viral CAP protein was very critical in terms of the activation of both AKT and HDAC6. Finally, the gC1qR binding activity of CAP was required for its innate immune modulatory function. The whole story is very complicated and not easy to follow. The study would be more convincing if the authors could refrain from drawing too many conclusions with relatively weak evidence.

**Part II – Major Issues: Key Experiments Required for Acceptance**

Reviewer #1: PCV2 Cap protein was identified as the viral protein that causes cGAS phosphorylation and polyubiquitination, this was supported by the data of experiments using PCV2 bearing mutated Cap protein. To demonstrate the direct role of Cap protein, it should be tested whether expression of Cap protein alone is able to inhibit IFN-beta production through causing cGAS phosphorylation and polyubiquitination.

Reviewer #2: Major concerns:

(1) Akt phosphorylates cGAS (S305, equivalent to S278 in porcine cGAS) and inhibits its catalytic activity, as referenced by the manuscript, has been reported in 2015. On the other hand, K48-linked ubiquitination of K48 has been reported to be a recognition signal for p62-dependent selective autophagic degradation (Chen et al. Mol Cell, 2016), with exactly the same lysine residue (K414 for hcGAS and K389 for porcine cGAS in this manuscript). The fact that the authors failed to cite and discuss their findings with the 2016 paper is concerning.

- Firstly, the author should cite and discuss their result with the 2016 paper. Based on the 2016 paper, ISD treatment, HSV-1 infection will induce the cGAS autophagic degradation in a manner similar to the EBSS-treated condition. In this manuscript, the authors propose that HDAC6-faciliated degradation is independent of EBSS-mediated cGAS degradation. If HDAC6 recognizes K48-PolyUb, what prevents it from interacting with EBSS-mediated cGAS ubiquitination (with same Lysine)? Please provide explanation and hypothesis in the discussion.

- Figure S4B shows that HDAC6 knockdown does not increase cGAS ubiquitination in the mock control group, while Figure S4C shows it increases the basal level of cGAS-Ub in the absence of viral infection. This needs to be repeated, since the increase of cGAS-Ub upon HDAC6 KD in mock-infected group might suggest that HDAC6 is also involved in starvation-mediated cGAS ubiquitination and degradation.

- Figure S4D needs quantification of the cGAS signal. To me it looks like knockdown of HDAC6 prevents cGAS from being degraded within 48 hours while the author claims that HDAC6 did not affect the degradation of cGAS in EBSS-treated cells.

(2) Figure 1A-1D, details on the number of pigs infected (n=?) needs to be provided, either on the figure or in the figure legend. Please also provide details on the experiment to obtain viral loads (which tissue? qRT-PCR?) in the materials and methods for Figure 1C. For 1C and 1D, the data were presented as “relative (%)”, the authors need to clarify how data are calculated and normalized.

(3) Please provide a coIP experiment to test the affinity of WT and K389R cGAS to p62.

(4) Please provide a control showing the levels of HDAC6 in Figure 4H.

(5) Figure 5H, cells reconstituted with cGAS-S278A seems to have comparable levels of cGAMP in mock and PCV2 infection. However, PCV2 should still trigger the degradation of cGAS-S278A, at least partially based on Figure 6. How does the author explain this? Please provide a WB showing the expression of these reconstituted cGAS in mock and infected group.

(6) Figure 6D, inhibitor of the AMPK partially saves cGAS from degradation. Could AMPK be the Akt-equivalent kinase in the EBSS-mediated cGAS degradation? Please provide thoughts on this (doesn’t need to be included in the manuscript)

(7) Figure 8B, please do a control infection with WT and RmA virus in HDAC6 knockdown cells + Akt inhibitor, and monitor IFN-b mRNA levels, to test whether the effect of the elevated IFN-b is solely due to the rescue of cGAS level and activity.

(8) Provide a mock-infected Brain and Lung IHC for Figure S7.

Reviewer #3: 1. Line 81-87. This part is somewhat over-exaggerated. The findings in this manuscript are based on PCV infection, which may not be generalized.

2. Figure 3D, cGAS was still degraded with Baf-A1 treatment, why?

3. Line 183, ‘although’ typo.

4. Line 225, the interpretation for fig 4E is not very accurate. K389R mutant was still ubiquitinated but it did not interact with HDAC6.

5. Figure 4F and G, the immunoprecipitated bands of p62 are kind of dubious.

6. Regarding p62 recruitment, previous study reveals that K414 ubiquitination of cGAS helps to recruit p62 (PMID: 27666593), which is contradictory to current study. Authors may consider providing some experimental evidence or discussion at least.

7. Figure 4H and I, it is not very convincing to conclude the necessity of deacetylase activity solely depending on an inhibitor, considering the potential off-target effect.

8. Figure5C and D, it seems that the production of cGAMP was not affected by PCV, why?

9. For figure 5 and 6, the authors heavily relied on an Akt inhibitor and a p-substrate antibody. It would be more convincing to provide some genetic evidence.

10. Figure 7, what is the role of gC1qR in cGAS activation in general? How is gC1qR relayed to AKT and PKCδ? Similarly, it would be more convincing if authors could provide some genetic evidence other than relying on a PKCδ inhibitor.

11. For figure 8, it would be critical to show that the mutant virus didn’t recruit gC1qR and gC1qR deficiency could abrogate the immune modulatory effect of PCV infection.

12. The discussion part needs to be significantly improved. The logical flow between paragraphs are vague.

**Part III – Minor Issues: Editorial and Data Presentation Modifications**

Reviewer #1: 1. Data in Western blots need to be quantified from at least three independent experiments. The differences that were claimed in some of the Western blots are very moderate, which need to be supported by quantification.

2. Please indicate the source and validation of the antibody that recognizes phosphorylated S278.

3. Line 33: it is not immediately clear how this study helps vaccine design.

4. What is level of IFN beta in PCV2-infected cells or piglets, without co-infection by other viruses?

5. Line 137: if the inactivated PCV2 can also lead to cGAS degradation, does PCV2 carry enough Cap protein to target and block cGAS function?

6. Line 424: levels of IFN beta in the media need to be directly measured by ELISA.

7. Figure 1 and also in other figures: please indicate the number of piglets used in each experiment.

8. Figure 1F: what is the baseline for the quantification of IFN beta mRNA?

9. Figure 3H, 4D, 6G: please present the PCV2 protein signal. Also, please quantify protein co-localization.

10. Figure 6D, S6E: blocking AMPk also increases cGAS level, similar to blocking Akt. Please discuss.

11. The language needs to be edited.

Reviewer #2: Minor:

(1) Define “EBSS” in the abstract (Line 24)

(2) Please provide background on cGAS Polyubiquitination studies before directly mentioning the ‘spatially transportation of ….” (Line 81)

(3) Provide more background on PCV2 in the introduction part, (e.g., DNA or RNA virus, size, life cycle, etc.)

(4) Again define “EBSS” in the result part (Line 178)

(5) Line 183, “although” should be deleted

(6) Line 187, how do you narrow down the region to only include 8 Ub sites? More details need to be given to justify the logic here.

(7) Line 201-205 needs revision. I understand that the author wants to propose a different model for PCV2-mediated cGAS autophagy. However, the K48 Ub and the site is shared between the EBSS and PCV2. In this case, the author cannot claim that K48 Ub chains serves as a signal only for the selective autophagy by PCV2 (Line 205).

(8) Line 416, why is Figure S7C described here? The whole section is on PCV2RmA mutant virus.

Reviewer #3: (No Response)

PLOS authors have the option to publish the peer review history of their article (what does this mean?). If published, this will include your full peer review and any attached files.

Reviewer #1: No

Reviewer #2: No

Reviewer #3: No
---

## [Decision Letter · Decision Letter 1]

18 Aug 2021

Dear Dr. Huang,

Thank you very much for submitting your manuscript "PCV2 targets cGAS to inhibit type I interferon induction to promote other DNA virus infection" for consideration at PLOS Pathogens. As with all papers reviewed by the journal, your manuscript was reviewed by members of the editorial board and by several independent reviewers. The reviewers appreciated the attention to an important topic. Based on the reviews, we are likely to accept this manuscript for publication, providing that you modify the manuscript according to the review recommendations.

Sincerely,

Pinghui Feng

Associate Editor

PLOS Pathogens

Blossom Damania

Section Editor

PLOS Pathogens

Kasturi Haldar

Editor-in-Chief

PLOS Pathogens

orcid.org/0000-0001-5065-158X

Michael Malim

Editor-in-Chief

PLOS Pathogens

orcid.org/0000-0002-7699-2064

Reviewer Comments (if any, and for reference):

Reviewer's Responses to Questions

**Part I - Summary**

Reviewer #1: The authors have adequately answered my questions.

Reviewer #2: The authors have adequately addressed all my concerns.

Some minor suggestions:

Line 580-588. Please edit the language to make it concise and accurate. For example, “PCV2 infection, but not EBSS treatment, activates HDAC6 to facilitate cGAS transportation, leading to an increased interaction between K48-ubiquitinated cGAS with p62”. “Inactivated HDAC6 may interact with K48-Ub cGAS but is not able to transport it.”

Manuscript still needs further revision for language.

Reviewer #3: Authors have done an excellent job to address my concerns and the manuscript is significantly improved. I have no more comments about the manuscript except some minor issues listed below:

1. Line 73. Reference #9 is incorrect, and the sentence needs to be rephrased. PMID: 34015248 and PMID: 30092200 should be referenced.

2. The writing still needs to be improved. For instance, line 83 ‘through RNF185 mediation’; line 87 ‘despite some molecules that’; line 91-92, grammar; line 99, deplete ‘as most people known’. The list is by no means exhaustive.

3. Line 247, the reconstitution experiments with HDAC6 WT and mutant were not described.

**Part II – Major Issues: Key Experiments Required for Acceptance**

Reviewer #1: No more major issues.

Reviewer #2: (No Response)

Reviewer #3: (No Response)

**Part III – Minor Issues: Editorial and Data Presentation Modifications**

Reviewer #1: No more minor issues.

Reviewer #2: (No Response)

Reviewer #3: (No Response)

PLOS authors have the option to publish the peer review history of their article (what does this mean?). If published, this will include your full peer review and any attached files.

Reviewer #1: No

Reviewer #2: No

Reviewer #3: No

Figure Files:

Data Requirements:

Reproducibility:

References:

---

## [Editor Report · Decision Letter 2]

3 Sep 2021

Dear Huang,

We are pleased to inform you that your manuscript 'PCV2 targets cGAS to inhibit type I interferon induction to promote other DNA virus infection' has been provisionally accepted for publication in PLOS Pathogens.

I would also like to point a minor change to make in this sentence, an increased interaction between K48-ubiquitinated cGAS with (should be and) p62.

Please note that your manuscript will not be scheduled for publication until you have made the required changes, so a swift response is appreciated.IMPORTANT: The editorial review process is now complete. PLOS will only permit corrections to spelling, formatting or significant scientific errors from this point onwards. Requests for major changes, or any which affect the scientific understanding of your work, will cause delays to the publication date of your manuscript.

Best regards,

Pinghui Feng

Associate Editor

PLOS Pathogens

Blossom Damania

Section Editor

PLOS Pathogens

Kasturi Haldar

Editor-in-Chief

PLOS Pathogens

orcid.org/0000-0001-5065-158X

Michael Malim

Editor-in-Chief

PLOS Pathogens

orcid.org/0000-0002-7699-2064
---

## [Editor Report · Acceptance letter]

15 Sep 2021

Dear prof Huang,

We are delighted to inform you that your manuscript, "PCV2 targets cGAS to inhibit type I interferon induction to promote other DNA virus infection," has been formally accepted for publication in PLOS Pathogens.

Best regards,

Kasturi Haldar

Editor-in-Chief

PLOS Pathogens

orcid.org/0000-0001-5065-158X

Michael Malim

Editor-in-Chief

PLOS Pathogens

orcid.org/0000-0002-7699-2064